# TreeLoRA: Efficient Continual Learning via Layer-Wise LoRAs Guided by a Hierarchical Gradient-Similarity Tree

**Yu-Yang Qian** [1 2]  **Yuan-Ze Xu** [1 2]  **Zhen-Yu Zhang** [3]  **Peng Zhao** [1 2]  **Zhi-Hua Zhou** [1 2]

## Abstract

Many real-world applications collect data in a streaming environment, where learning tasks are encountered sequentially. This necessitates *continual learning* (CL) to update models online, enabling adaptation to new tasks while preserving past knowledge to prevent catastrophic forgetting. Nowadays, with the flourish of *large pre-trained models* (LPMs), *efficiency* has become increasingly critical for CL, due to their substantial computational demands and growing parameter sizes. In this paper, we introduce TreeLoRA (K-D Tree of Low-Rank Adapters), a novel approach that constructs *layer-wise* adapters by leveraging hierarchical gradient similarity to enable efficient CL, particularly for LPMs. To reduce the computational burden of task similarity estimation, we employ *bandit* techniques to develop an algorithm based on lower confidence bounds to efficiently explore the task structure. Furthermore, we use sparse gradient updates to facilitate parameter optimization, making the approach better suited for LPMs. Theoretical analysis is provided to justify the rationale behind our approach, and experiments on both *vision transformers* (ViTs) and *large language models* (LLMs) demonstrate the effectiveness and efficiency of our approach across various domains, including vision and natural language processing tasks.

## 1. Introduction

Machine learning algorithms have achieved significant success across a wide range of applications. However, distribution shift still poses a challenge to machine learning algorithms (Bengio et al., 2021; Zhou, 2022; Sutton et al., 2022), where new tasks with different data distributions are encountered sequentially, leading to a performance degradation over time (Vela et al., 2022). This sequential nature necessitates a learning method capable of continuous adaptation and evolution, named *continual learning* (CL), in which models must adapt to new tasks while retaining knowledge from previous ones to alleviate catastrophic forgetting (Thrun & Pratt, 1997).

Existing CL approaches can be generally classified into three main categories (Wang et al., 2024b): regularization-based methods (Kirkpatrick et al., 2017), which introduce specially designed regularizers to prevent the model from forgetting previously learned knowledge; rehearsal-based methods (Shin et al., 2017), which store data from previous tasks in a buffer and reuse it during the ongoing task stream; and architecture-based methods (Mallya & Lazebnik, 2018; Serrà et al., 2018), which design specialized model architectures to dynamically adapt to new tasks and mitigate forgetting. These approaches have demonstrated strong empirical performance for continual learning tasks with conventional machine learning models.

Nowadays, with the rise of the pre-training paradigm, *large pre-trained models* (LPMs) have excelled in numerous tasks such as natural language processing (Brown et al., 2020) and computer vision (Radford et al., 2021). The necessity of *efficient CL*, especially the approaches that are tailored to LPMs, has become increasingly demanding, due to the substantial computational demands and growing parameter sizes. Therefore, it is essential to explore the issue of efficient continual learning, particularly the *Continual Fine-Tuning* (CFT) of LPMs. Although there exist some continual learning methods developed for LPMs (Scialom et al., 2022; Wang et al., 2023b; Dou et al., 2024), they often do not focus on the efficiency in CL, typically exhibiting computational complexity that scales *linearly* with the number of tasks. Consequently, this raises a critical new challenge: *How to develop an efficient continual learning approach, particularly fitted for large pre-trained models?*

In this paper, we propose an efficient continual learning approach that facilitates adaptation to new tasks by leveraging task-shared knowledge, while avoiding the linear computa-

[1]National Key Laboratory for Novel Software Technology, Nanjing University, China [2]School of Artificial Intelligence, Nanjing University, China [3]RIKEN AIP, Tokyo, Japan. Correspondence to: Peng Zhao <zhaop@lamda.nju.edu.cn>.

*Proceedings of the $42^{nd}$ International Conference on Machine Learning*, Vancouver, Canada. PMLR 267, 2025. Copyright 2025 by the author(s).

tional complexity that scales with the number of tasks. Our approach, TreeLoRA (short for "K-D Tree of Low-Rank Adapters"), introduces layer-wise adapters to construct a *hierarchical tree* based on the criteria of gradient similarity. This design organizes and manages historical tasks efficiently, thereby eliminating linear task-dependence. To ensure the overall efficiency, two key components are developed. First, it is necessary to estimate the gradient similarity on the fly during training, and we employ *bandit* techniques and develop an algorithm based on the lower confidence bound (Garivier & Moulines, 2011) to efficiently search on the tree structure. Second, to better suit the LPMs, we apply sparse gradient updates to optimize model parameters more efficiently for new task adaptation.

Our approach shows advantages both in theory and practice. Theoretically, owing to our designed tree structure, our approach achieves tighter regret bounds compared to conventional bandit search methods that do not utilize such a structure, justifying the rationale behind our approach. Empirically, we conduct experiments on vision transformers and large language models, across both the computer vision benchmarks and natural language processing tasks, to validate the performance and efficiency of our approach. The experimental results demonstrate that, with similar or even improved performance, our method achieves up to a 3.2× speedup for ViTs and 2.4× speedup for LLMs in training time, compared to previous state-of-the-art methods, thereby validating the effectiveness and efficiency of our approach.

**Organization.** Section 2 introduces the problem formulation. Section 3 presents our TreeLoRA approach. Section 4 provides theoretical guarantees. Section 5 reports experiments, and Section 6 concludes the paper. Due to page limits, we defer more empirical studies to Appendix A and related works to Appendix B. All proofs are in Appendix C.

## 2. Problem Formulation

In this section, we introduce the problem formulation. Our setting is a typical continual learning problem (Lange et al., 2022), where there are two stages: (i) the offline initialization stage and (ii) the online adaptation stage.

(i) In the offline initialization stage, the learner has access to a lot of labeled data sampled from initial distribution $\mathcal{D}_0$ defined over the feature space $\mathcal{X}$ and label space $\mathcal{Y}$, and can use these data to learn a good pre-trained model parameter $\mathbf{w}_0 \in \mathbb{R}^d$.

(ii) In the online adaptation stage, a sequence of tasks $\{\mathcal{T}_1, \mathcal{T}_2, \ldots, \mathcal{T}_N\}$ arrive sequentially, where $N$ is the total number. For the $n$-th task, the learner receives a few labeled data $S_n = \{\mathbf{x}_n^t, y_n^t\}_{t=1}^{m_n}$ sampled from distribution $\mathcal{D}_n$. The learner needs to use these data to update the current model parameter $\mathbf{w}_n \in \mathbb{R}^d$.

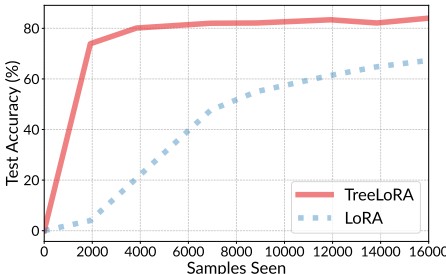

**Figure 1:** Test accuracy vs. the number of arrived samples for our *TreeLoRA* and the vanilla *LoRA* (Hu et al., 2022) on the Split CIFAR-100 dataset when encountering a new task, demonstrating our efficiency to adapt to new tasks, owing to mining the hierarchical gradient-similarity structure to explore task-shared knowledge.

The goal is to learn a good model that performs well not only on the current task but also on previous ones. Specifically, we denote $f_i(\mathbf{w}_j)$ as the prediction performance (such as the accuracy, SARI score, and ROUGE-L score, etc.) evaluated on task $i$ using the parameters updated up to task $j$, then the *overall performance* (OP) for the $n$-th model is

$$\text{OP}_n \triangleq \frac{1}{n} \sum_{i=1}^{n} f_i(\mathbf{w}_n).$$

Another important goal is to minimize the *forgetting* rate, which is defined as *backward transfer* (BWT) that measures catastrophic forgetting by comparing a model's performance on old tasks before and after learning new ones, that is

$$\text{BWT}_n \triangleq \frac{1}{n} \sum_{i=1}^{n} \left( f_i(\mathbf{w}_i) - f_i(\mathbf{w}_n) \right),$$

which measures the performance degradation on previous tasks. Therefore, the goal is to learn a model $\mathbf{w}_n$ that can achieve *maximal* OP and *minimal* BWT. Note that we focus on a more challenging task where the identity of the task is not available during testing.

## 3. Our Approach

This section presents our approach. First, we introduce our novel continual update framework for large pre-trained models, leveraging LoRAs with a tree structure. Next, we detail the construction of the tree using a bandit algorithm. Finally, we explain how the model is updated efficiently through sparse updates facilitated by the TreeLoRA structure.

### 3.1. Constructing Task-Similarity Tree Structure

In this part, we will detail our novel continual update framework for LPMs, leveraging LoRAs with a tree structure.

Our key technical insight is that some tasks exhibit similar characteristics and can be grouped together, allowing for

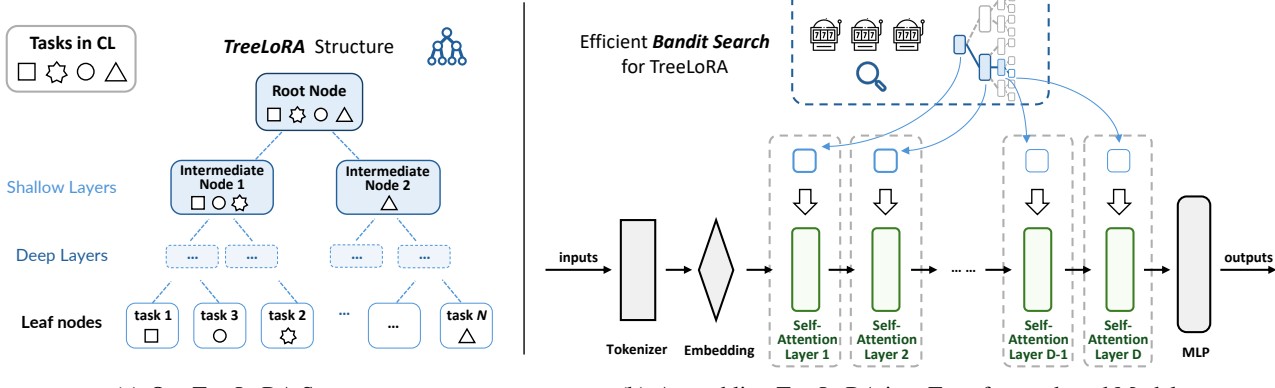

(a) Our TreeLoRA Structure          (b) Assembling TreeLoRA into Transformer-based Models

**Figure 2:** Figure illustrating our approach: (a) We develop the TreeLoRA approach for efficient continual learning, where each node represents a task group, and each leaf node corresponds to an individual task. The children of each node represent subgroups that belong to the parent node. (b) Employing TreeLoRA for transformer-based models: when encountering a new task, we first perform the bandit algorithm to identify the most relevant task group, then assemble the corresponding LoRA adapter to efficiently adapt to the task.

the use of task-shared knowledge to improve performance and accelerate adaptation to new tasks. Conversely, tasks that differ significantly or even conflict with others are handled separately to alleviate catastrophic forgetting. With this insight, we propose a novel approach that explicitly leverages the *hierarchical task-similarity structure* to enhance the efficiency of continual learning. Specifically, we introduce *TreeLoRA*, designed to explore and manage task relationships, whose effectiveness is shown in Figure 1.

**Tree Structure Design.** Consider a deep neural network as the learning model, which can be divided into multiple layers of parameters and connected by activation functions. Shallow layers (closer to the input) tend to capture task-shared common patterns, while the deeper layers (closer to the output) capture task-specific semantics. Our TreeLoRA is designed to mirror this structure, as illustrated in Figure 2: the root node represents all tasks, and nodes in shallow layers represent adapters that capture shared patterns across several task groups. Nodes in deeper layers represent task-specific semantics, and the leaf nodes correspond to adapters for individual tasks.

**Task Similarity Measurement.** We organize tasks based on the similarity of the gradient directions – tasks with similar gradient directions are grouped together, facilitating shared knowledge extraction and streamlined continual learning. Formally, consider a data stream of $N$ tasks. At round $n \in [N]$, we receive a new task $n$ associated with a labeled dataset $S_n = (\mathbf{x}_n^t, y_n^t)_{t=1}^{m_n}$ sampled from the task distribution $\mathcal{D}_n$. The expected gradient direction for task $n$ of model $k$ (w.r.t. the task $\mathcal{T}_k$) is defined as:

$$\mathbf{g}_k \triangleq \mathbb{E}_{(\mathbf{x},y)\sim\mathcal{D}_n} \left[ \nabla \ell(\mathbf{w}_k; \mathbf{x}, y) \right],$$

and the TreeLoRA structure is constructed based on the similarity of gradient directions. Specifically, the node $\mathcal{N}_j$ (representing a set of tasks) is defined as:

$$\mathcal{N}_j = \max\{\mathcal{N} \subseteq [N] : \|\mathbf{g}_i - \mathbf{g}_{i'}\|_1 \leq \delta, \forall i, i' \in \mathcal{N}\}, \quad (1)$$

where $\delta$ is the similarity threshold. Note that we choose L1-norm as the similarity measure, as L1-norm is *more robust* in high-dimensional spaces (Aggarwal et al., 2001). This hierarchical task structure facilitates efficient adaptation to new tasks while preserving knowledge from previous tasks. By grouping tasks with similar characteristics, shared knowledge can be leveraged to accelerate learning and enhance continual learning performance.

To summarize, our TreeLoRA structure begins with a root node and incrementally builds a hierarchical tree structure as new tasks arrive. For each incoming task, it utilizes a bandit algorithm to identify the most suitable branch that shares similar gradient directions. Sparse gradient updates are then performed to efficiently adapt the model parameters while retaining previously acquired knowledge. At the end of each task stream, the key parameters of that task are recorded, and the TreeLoRA structure is updated accordingly. The complete algorithm is outlined in Algorithm 1. In the following sections, we provide a detailed explanation of how the tree is adaptively constructed using the bandit algorithm, and our sparse updates mechanism.

### 3.2. Adaptive Tree Construction via Bandit Algorithm

In this section, we demonstrate how the tree structure is constructed. As illustrated in the previous section, TreeLoRA captures task similarity through gradient direction. A significant challenge, however, lies in the fact that the gradient direction can only be determined after calculating gradients of

*all* previous tasks with *linear* complexity. This requirement makes it computationally infeasible, as it would require performing gradient descent $\mathcal{O}(N)$ times, for all previous tasks. To overcome this limitation, we introduce a novel *bandit* algorithm that efficiently constructs the tree structure on the fly with only *one* gradient query each round, therefore greatly improving the computational efficiency.

**Converting to Multi-Armed Bandits Problem.** The key dilemma lies in the costly computation complexity of calculating all previous gradients to determine the gradient similarity between the current task and previous tasks. To tackle this challenge, instead of calculating all previous gradients, we only select one most *promising* previous task to query the gradient and estimate the similarity. This scenario naturally aligns with an exploration-exploitation trade-off. Specifically, we convert the challenge to *multi-armed bandits* (MAB) problem, a fundamental problem in the online learning field, where the learner does not have access to all previous gradients but can only selectively obtain some (or even just one) gradient from the set of previous tasks.

Formally, we treat each previous task $k$ ($k \in [n-1]$) as a bandit arm. The gradient of the current sample is denoted as $\widehat{\mathbf{g}}_n^t \triangleq \nabla \ell(\mathbf{w}_n; \mathbf{x}_n^t, y_n^t)$, $t \in \{1, \ldots, m_n\}$, and the previous gradient (w.r.t. that tree branch) is $\widehat{\mathbf{g}}_k^t \triangleq \nabla \ell(\mathbf{w}_k; \mathbf{x}_n^t, y_n^t)$. At each iteration $t \in \{1, \ldots, m_n\}$, the learner can only pull one arm, i.e., choose only one previous task $k \in [n-1]$, and obtain the corresponding *empirical residual gradient* $\hat{\xi}_t^k$, which is the sum of the empirical gradients' differences

$$\hat{\xi}_t^k \triangleq \sum_{i=1}^{d'} \mathcal{S}\big[\nabla \ell(\mathbf{w}_n; \mathbf{x}_n^t, y_n^t) - \nabla \ell(\mathbf{w}_k; \mathbf{x}_n^t, y_n^t)\big]_i,$$

where $\mathcal{S}[\cdot]$ is the low-rank sparsify function that updates only the most relevant parameters, i.e., the task-specific low-rank adapter. Here, $d'$ represents a reduced dimensionality such that $d' \ll d$. $\hat{\xi}_t^k$ represents the estimated difference between the gradient of the current sample and previous task-shared gradient. The true target $\xi^{k^\star}$, however, is nearest task's *expected residual gradient*, defined as

$$\xi^{k^\star} \triangleq \sum_{i=1}^{d} \mathbb{E}_{(\mathbf{x},y) \sim \mathcal{D}_i} \big[\nabla \ell(\mathbf{w}_n; \mathbf{x}, y) - \nabla \ell(\mathbf{w}_{k^\star}; \mathbf{x}, y)\big]_i,$$

where $k^\star$ is the most relevant task for the current task $i$. The goal is to design a bandit algorithm to minimize cumulative loss between the selected arm and the optimal arm.

Inspired by previous bandit algorithms (Coquelin & Munos, 2007), we introduce *Lower Confidence Bound* (LCB) algorithm to adaptively construct the tree structure. Specifically, the LCB of the $k$-th node in the TreeLoRA is defined as

$$\text{LCB}_k = \begin{cases} \widehat{\mu}_k - 2\sqrt{\frac{\log t}{n_k}}, & \text{if } k \in \mathcal{L} \\ \max\left\{\min_{j \in \mathcal{C}} \left\{\widehat{\mu}_j - 2\sqrt{\frac{\log t}{n_j}} - \delta\right\}\right\}, & \text{if } k \notin \mathcal{L} \end{cases} \quad (2)$$

**Algorithm 1** TreeLoRA for Efficient Continual Learning
***
**Input:** learning rate $\alpha$, regularization coefficient $\lambda$
**Initialize:** task similarity estimation $\widehat{\mu}_k = 0, \forall k \in [N]$
Initialize the TreeLoRA structure with a root node;
**for** $n = 1, \ldots, N$ **do**
    Receive $n$-th task with labeled data $S_n = \{(\mathbf{x}_n^t, y_n^t)\}_{t=1}^{m_n}$;
    **for** $t = 1, \ldots, m_n$ **do**
        Use LCB to select most promising branch as Eq. (2);
        Employ sparse gradient updates as Eq. (3);
        Update the estimated task similarity $\widehat{\mu}_k$.
    **end**
    Record the task-specific adapter for the $n$-th task;
    Update the TreeLoRA structure as in Section 3.3.
**end**
***

where $\widehat{\mu}_k = \frac{1}{|\text{Select}_k|} \sum_{\tau \in \{\text{Select}_k\}} \hat{\xi}_\tau^k$ is the estimated task similarity between the current task and the $k$-th task group (i.e., the nodes in the branch of the selected leaf node at round $t$), $\mathcal{L}$ is the set of all leaf nodes, and $\mathcal{C}$ is the child nodes of the $k$-th node. The term $2\sqrt{\log t / n_k}$ in the LCB accounts for the uncertainty in the similarity estimation, encouraging exploration of less frequently selected nodes.

The LCB is computed for each layer of the model. By calculating the LCB layer by layer, TreeLoRA captures the similarity between tasks at various levels throughout the model hierarchy as illustrated in Figure 2, allowing us to better capture hierarchical task similarities, which is especially advantageous in transformer-based models. Our LCB algorithm determines the most promising task group for the current task by selecting the node $k$ with the minimum LCB value. This approach balances the trade-off between exploring potential task similarity relationships and exploiting known, promising task groups to optimize the tree construction process.

**Bandit Algorithm on a Smooth Tree.** By leveraging the hierarchical structure of TreeLoRA, our bandit algorithm on a smooth tree (Coquelin & Munos, 2007) significantly enhances efficiency compared to conventional bandit algorithms that do not utilize this structure. As stated in Theorem 1, our algorithm reduces the regret from the conventional $\mathcal{O}(\sqrt{N})$ to $\mathcal{O}(\sqrt{\log N})$ w.r.t. the number of tasks $N$. Consequently, we achieve faster identification of the most similar task group, which substantially reduces computational complexity. This improvement demonstrates the advantage of TreeLoRA to efficiently search for relevant tasks and mine the hierarchical task structure.

### 3.3. Efficient Sparse Updates

In this part, we will detail how to efficiently update the model by leveraging the hierarchical task structure. Specif-

ically, we only sparsely update a portion of the model parameters based on the hierarchical task structure. As shown in Figure 3, we first identify the most relevant task group for the current task based on our bandit algorithm. We then perform a sparse update on the model parameters, where only the parameters that are relevant are updated.

Specifically, for the $n$-th task, we choose the most relevant parameters by an adaptive regularization term as the regularizer, which is formally defined as

$$\ell_{\text{reg}}^t \triangleq |\hat{\xi}_t^k| = \|\hat{\mathbf{g}}_n^t - \hat{\mathbf{g}}_k^t\|_1.$$

This regularizer term enables adaptively selective parameter updates for the current task group by identifying and modifying only the most relevant parameters while preserving previously learned knowledge. The final loss combines the task loss and the regularizer term, leading to the formal model update definition as follows

$$\mathbf{w}_n^{t+1} = \mathbf{w}_n^t - \alpha \cdot \mathcal{S}\left[\nabla \ell(\mathbf{w}_n^t; \mathbf{x}_t, y_t)\right] - \lambda \cdot \nabla \ell_{\text{reg}}^t, \quad (3)$$

where $\alpha$ is the learning rate, $\mathcal{S}$ is the low-rank sparsify function, and $\lambda$ denotes the regularization coefficient. Finally, the updated model is obtained based on the previously learned model parameters and assembling the newly learned sparse update part, as illustrated in Figure 3.

This sparse update strategy enables us to efficiently adapt the model to new tasks while preserving knowledge of previously learned ones. At the end of each task stream, we record the key parameters for the $n$-th task, which is the set of relevant parameters that are updated for the current task, and we add these newly-obtained sparse parameters into the TreeLoRA structure and update the tree structure accordingly, as demonstrated in Algorithm 1.

**Practical Implementation.** In practice, we draw inspiration from LoRA (Hu et al., 2022) to perform the sparse update, a widely used method for efficiently tuning LPMs. Specifically, we utilize LoRA adapters to approximate the gradient of the current task and then apply sparse updates to model parameters. For the ViTs, we set the tree depth to 5; and for LLMs, we set it to 64. We clarify that tree depth is not directly determined by the number of tasks, as a single node can contain multiple tasks, allowing it to scale to a large number of tasks. However, the depth should not exceed the number of transformer layers, and it is treated as a tunable hyperparameter. Further empirical analysis of tree depth is provided in Section 5.2.

Inspired by the K-D tree data structure (Bentley, 1990), the task-similarity threshold $\delta$ does not require manual tuning, as it is automatically determined when we construct the tree after each task. Specifically, at each split, the threshold is computed by taking the median of the similarity, i.e., L1-norm, between each task gradient and the mean gradient

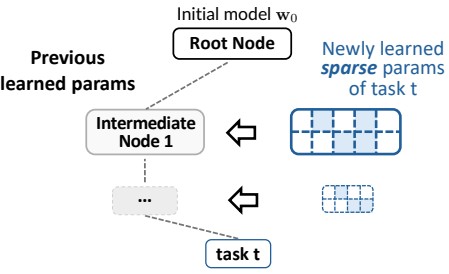

**Figure 3:** Illustration of the sparse update mechanism, where the blue part is the sparse update part we obtained for the $t$-th task.

within the corresponding task group, ensuring balanced tree growth and adaptive partitioning. After each task, we store the task-specific LoRA adapter and update the tree by inserting adapter into the leaf node of nearest branch by a depth-first search, thus adding new nodes and expanding the tree. If nodes exceed storage budget, we choose the closest adapters and reduce them to a single one. After the training, we select the most recently learned adapter for inference.

**Remark 1** (Discussion of Efficiency). The hierarchical tree structure and sparse update strategy significantly reduce the computational complexity and storage of the algorithm. Previous methods typically require computing gradients of all previous tasks, resulting in a complexity of $\mathcal{O}(N)$ per round. In contrast, our tree-based bandit algorithm avoids this linear computational complexity which scales with the number of tasks, but only explores the relevant tree branches at each round and achieves a complexity of $\mathcal{O}(1)$. The storage complexity is also minimal, as we store only the low-rank adapter for each task, which is much smaller than the full model parameters. Furthermore, our method requires only constant GPU memory during training, thanks to the efficiency of the bandit search strategy.

**Remark 2** (Generality of TreeLoRA). Our TreeLoRA structure is broadly applicable to various machine learning models, including deep neural networks, random forests, and other architectures. While applicable to a wide range of model architectures, it provides particular benefits for *transformer-based models* due to their large parameter count and inherent hierarchical structure. The layered organization of transformers naturally aligns with TreeLoRA's hierarchical task structure, enabling efficient knowledge sharing and task adaptation. This makes TreeLoRA especially effective for continual learning with transformer models, though its benefits extend to other architectures as well.

## 4. Theoretical Analysis

In this section, we provide theoretical guarantees for our TreeLoRA. We first introduce the measure of pseudo-regret, then provide regret bound analysis of our algorithm.

As demonstrated in Section 3.2, we convert the task structure

searching to a bandit problem. In this section, we provide the theoretical guarantee of regret bound for our algorithm. The measure is the *pseudo-regret* (Bubeck & Cesa-Bianchi, 2012), which is the difference of cumulative loss between the selected arm and the optimal arm, formally,

$$\mathbf{Reg}(T) \triangleq \mathbb{E}\left[\sum_{t=1}^{T} \hat{\xi}_t^k\right] - \min_{k^\star \in [n-1]} \sum_{t=1}^{T} \xi^{k^\star},$$

where $T$ is the total number of iterations. To establish theoretical guarantees for our bandit algorithm on the tree structure, note that our TreeLoRA naturally groups similar tasks into the same branch, enabling efficient mining and managing of task similarity structure. We formalize this observation of smooth tree as Assumption 1.

**Assumption 1** (Assumption $A_\eta$ of Coquelin & Munos (2007)). *For any node $i \in \mathcal{I}_\eta$ in TreeLoRA with depth $d < D$, there exists $\delta_d > 0$, such that for all $j \in L(\mathcal{N}_i)$,*

$$|\xi^j - \xi^i| \le \delta_d,$$

*where $\delta_d$ is the smooth coefficient, $\mathcal{I}_\eta \triangleq \{\text{node } i, \Delta_i \le \eta\}$ is the set of $\eta$-optimal nodes, in which $\Delta_i \triangleq \xi^i - \xi^{k^\star}$ is the suboptimality gap. $L(\mathcal{N}_i)$ denotes the set of leaf nodes in the branch of node $\mathcal{N}_i$, and $\xi^i$ represents the residual gradient of task $i$, formally defined as $\xi^i \triangleq \mathbb{E}_{(\mathbf{x},y)\sim\mathcal{D}_n} [\nabla \ell(\mathbf{w}_n; \mathbf{x}, y) - \nabla \ell(\mathbf{w}_i; \mathbf{x}, y)].*

This assumption characterizes the nature smoothness property of our TreeLoRA structure, where tasks grouped within the same branch are inherently similar by constructing a tree as Eq. (1) such that the leaves of a branch have similar values. The smoother the tree, the smaller $\delta_d$ will be. Further, if we assume that the empirical residual gradient $\hat{\xi}_t^k$ has some unbiased noise that can be eliminated by the sparsify operation, then it indicates that $\mathbb{E}[\hat{\xi}_t^k] = \xi^k$, which means the learner can obtain an *unbiased estimation* of the true residual gradient at each iteration.

We are now ready to provide the following theoretical guarantee for our proposed algorithm.

**Theorem 1** (Regret Bound). *Under Assumption 1, assume $A_\eta$ with an exponential sequence $\delta_d = \delta\gamma^d$. Then, with probability at least $1 - 1/T$, the cumulative regret of TreeLoRA after $T$ rounds satisfies:*

$$\mathbf{Reg}(T) \le \mathcal{O}\left(\sqrt{T|J_\eta| \log\left(\frac{NT}{|J_\eta|}\right)} + \frac{\delta^c}{\eta^{2+c}} \log\left(\frac{NT}{\eta^2}\right)\right),$$

*where $J_\eta \triangleq \{\text{leaf node } i, |\xi^i - \xi^{k^\star}| \le \eta\}$ is the set of leaf nodes with a suboptimality gap less than $\eta$, $T$ is the number of rounds, $N$ is the number of tasks, and $c \triangleq \log(2)/\log(1/\gamma)$ is a constant.*

Note that $J_\eta$ is the set of leaf nodes with a suboptimality gap less than $\eta$, which implies that $|J_\eta| \le N$. Consequently, Theorem 1 demonstrates that the regret's dependency on the number of tasks improves from the conventional order of $\sqrt{N}$ to a problem-dependent order of $\sqrt{|J_\eta|}$. If $\delta = +\infty$ and $\eta = +\infty$, the smooth tree structure reduces to the trivial case of non-smoothness, resulting in $|J_\eta| = N$. In this scenario, we recover the conventional regret bound of the LCB algorithm without structural assumptions, i.e., $\mathcal{O}(\sqrt{TN \log T})$. On the other hand, in a benign case where $\delta$, $\eta$ and $|J_\eta|$ can be treated as constants, the regret bound simplifies to $\mathcal{O}(\sqrt{T \log(NT)})$, reducing the dependency on the number of tasks from $\mathcal{O}(\sqrt{N})$ to $\mathcal{O}(\log(N))$. Theorem 1 validates the rationality and effectiveness of our proposed bandit search strategy with the hierarchical tree structure. The proof of Theorem 1 is deferred to Appendix C.1.

## 5. Experiments

This section provides the experimental results of our approach. To comprehensively evaluate the effectiveness and efficiency of our proposed TreeLoRA, we conduct empirical experiments across two modern model architectures: *vision transformers* (ViT) and *large language models* (LLMs), across both the computer vision and the natural language processing tasks.[1] Our experimental evaluation aims to answer the following three research questions:

- **Q1:** How does TreeLoRA perform compared to existing continual learning methods?
- **Q2:** How efficient is our proposed TreeLoRA in terms of the computational efficiency?
- **Q3:** How effective is the TreeLoRA to capture the inherent task similarity structure?

### 5.1. Evaluation on Vision Transformers

We first evaluate on vision transformer models using three widely-adopted computer vision benchmarks, i.e., Split CIFAR-100, Split ImageNet-R, and Split CUB-200.

**Contenders.** To validate the effectiveness of our approach, we compared our TreeLoRA with several recent contenders. The following seven methods can be categorized into five groups: (i) The baseline method *SeqLoRA*, which is a standard LoRA method that naively learns a new LoRA adapter for each task; (ii) Rehearsal-based methods represented by *GEM* (Lopez-Paz & Ranzato, 2017), which utilize episodic memory to prevent catastrophic forgetting; (iii) Regularization-based methods exemplified by *EWC* from (Kirkpatrick et al., 2017), which constrain parameter updates to prevent forgetting, (iv) Prompt-based methods

---

[1]Our code and datasets are available at https://github.com/ZinYY/TreeLoRA

**Table 1:** Comparison of different continual learning methods on ViTs with three benchmark datasets. For each method, we report the average accuracy (ACC (%) ↑), backward transfer (BWT (%) ↓), and training time (TIME (s) ↓) metrics. Results are averaged over three runs with standard deviations. The best results are highlighted in bold.

| | SeqLoRA | GEM | EWC | L2P | DualPrompt | HiDePrompt | HiDeLoRA | TreeLoRA |
|---|---|---|---|---|---|---|---|---|
| | *Split CIFAR-100* | | | | | | | |
| ACC (%) ↑ | $10.35 \pm 0.35$ | $76.46 \pm 0.35$ | $84.79 \pm 0.09$ | $79.66 \pm 0.60$ | $78.83 \pm 0.53$ | $83.71 \pm 0.03$ | $88.46 \pm 0.04$ | $\mathbf{88.54 \pm 0.05}$ |
| BWT (%) ↓ | $97.62 \pm 0.11$ | $4.37 \pm 0.30$ | $5.22 \pm 0.04$ | $4.40 \pm 0.01$ | $4.89 \pm 1.67$ | $5.34 \pm 0.26$ | $4.33 \pm 0.41$ | $4.37 \pm 0.15$ |
| TIME (s) ↓ | $694 \pm 8$ | $22456 \pm 31$ | $4091 \pm 14$ | $1240 \pm 12$ | $1173 \pm 14$ | $1205 \pm 11$ | $692 \pm 7$ | $\mathbf{214 \pm 4}$ |
| | *Split ImageNet-R* | | | | | | | |
| ACC (%) ↑ | $8.65 \pm 0.15$ | $54.45 \pm 0.76$ | $55.43 \pm 0.54$ | $56.75 \pm 0.61$ | $54.81 \pm 0.40$ | $62.94 \pm 0.91$ | $\mathbf{72.28 \pm 0.15}$ | $71.94 \pm 0.47$ |
| BWT (%) ↓ | $82.55 \pm 0.93$ | $2.16 \pm 0.40$ | $9.10 \pm 0.97$ | $3.66 \pm 0.16$ | $4.39 \pm 0.11$ | $3.18 \pm 0.56$ | $4.16 \pm 0.05$ | $4.06 \pm 0.40$ |
| TIME (s) ↓ | $797 \pm 11$ | $17244 \pm 29$ | $4727 \pm 21$ | $1431 \pm 10$ | $1357 \pm 15$ | $1354 \pm 13$ | $796 \pm 9$ | $\mathbf{260 \pm 5}$ |
| | *Split CUB-200* | | | | | | | |
| ACC (%) ↑ | $9.10 \pm 0.41$ | $61.28 \pm 0.45$ | $36.34 \pm 0.34$ | $39.72 \pm 0.34$ | $37.52 \pm 0.37$ | $63.86 \pm 0.11$ | $73.48 \pm 1.35$ | $\mathbf{73.66 \pm 0.22}$ |
| BWT (%) ↓ | $83.45 \pm 0.23$ | $2.64 \pm 0.30$ | $6.65 \pm 0.53$ | $7.98 \pm 0.70$ | $11.91 \pm 0.04$ | $5.57 \pm 0.23$ | $5.38 \pm 0.21$ | $4.87 \pm 0.30$ |
| TIME (s) ↓ | $194 \pm 11$ | $7233 \pm 18$ | $1702 \pm 9$ | $333 \pm 5$ | $315 \pm 4$ | $307 \pm 1$ | $194 \pm 3$ | $\mathbf{86 \pm 3}$ |

(a) error curve

(b) efficiency comparison

(c) speed-accuracy comparison

**Figure 4:** (a) Comparison of performance on the Split CIFAR-100. (b) Comparison of the efficiency of different methods. (c) Comparison of accuracy and efficiency (with mean and standard deviation over three runs). The closer to the top-right corner, the better the algorithm.

including *L2P* (Wang et al., 2022b) and *DualPrompt* (Wang et al., 2022a), which learn prompts while keeping the model parameters frozen; and (v) the hierarchical decomposition methods including *HiDePrompt* (Wang et al., 2023a) and *HiDeLoRA* (Wang et al., 2024a), which divide the continual learning into sub-components. Further comparative results, including those of SAPT (Zhao et al., 2024b) and TASL (Feng et al., 2024), are provided in Appendix A.

**Results Analysis of ViTs Experiments.** To answer question **Q1**, we compare TreeLoRA with state-of-the-art methods across three benchmark datasets on ViTs. As shown in Table 1, the naive *SeqLoRA* does not perform well, as it learns a new adapter for each task, leading to catastrophic forgetting. Conventional CL methods, such as *GEM* and *EWC*, are not specifically designed for LPMs, and do not achieve the same level of performance as more recent methods. TreeLoRA outperforms previous state-of-the-art methods, *HiDeLoRA* and *HiDePrompt*, achieving the best accuracy on two out of three datasets. This superior performance can be attributed to TreeLoRA's ability to exploit task similarity structures and leverage shared knowledge across tasks.

**Efficiency Evaluation.** To answer question **Q2**, we present

**Table 2:** Performance comparison of TreeLoRA with different training epochs on Split CIFAR-100. Results show that TreeLoRA achieves good performance even with few training epochs.

| | Acc (%) ↑ | Bwt (%) ↓ | Time (s) ↓ |
|---|---|---|---|
| *TreeLoRA (2 epochs)* | $87.73 \pm 0.12$ | $4.13 \pm 0.25$ | $23 \pm 0.27$ |
| *TreeLoRA (10 epochs)* | $88.23 \pm 0.16$ | $4.29 \pm 0.11$ | $108 \pm 1.54$ |
| *TreeLoRA (20 epochs)* | $88.54 \pm 0.05$ | $4.37 \pm 0.15$ | $214 \pm 4.12$ |

the training time of different methods. As shown in Table 1 and Figure 4(b), our TreeLoRA achieves comparable performance compared to previous state-of-the-art methods, while requiring significantly less training time, resulting in a 3.2× speedup. Figure 4(c) illustrates the speed-accuracy trade-off of different methods, where methods closer to the top-right corner are better, as they achieve higher accuracy with less training time. Our TreeLoRA ranks among the top performers and processes nearly three times more samples per second than previous state-of-the-art methods.

We also compare TreeLoRA's performance with different training epochs. As shown in Table 2, TreeLoRA achieves a good performance even with just two epochs of training, reaching 87.73% accuracy on the Split CIFAR-100.

**Table 3:** Comparison of different continual learning methods on various foundation models on the TRACE dataset (Wang et al., 2023c). For each method, we report the overall performance (OP (%) ↑), backward transfer (BWT (%) ↓), and training time (TIME (s) ↓) metrics. Results are averaged over three runs with standard deviations. The best results are highlighted in bold.

| | FIX (ICL) | SeqLoRA | OGD | GEM | EWC | L2P | DualPrompt | HiDeLoRA | O-LoRA | TreeLoRA |
|---|---|---|---|---|---|---|---|---|---|---|
| | | | | | *mistralai / Mistral-7B-Instruct-v0.3* | | | | | |
| OP (%) ↑ | 45.04±0.3 | 46.94±1.2 | 45.44±1.6 | 52.32±1.2 | 52.45±1.3 | 49.32±0.8 | 51.14±1.2 | 51.81±0.9 | 52.02±0.8 | **54.77±1.1** |
| BWT (%) ↓ | — | 11.41±0.6 | 12.75±0.7 | 6.01±0.3 | 5.98±0.8 | 5.34±0.6 | 6.13±0.5 | 6.25±0.3 | 8.13±0.6 | 3.77±0.4 |
| TIME (s) ↓ | — | 1091±25 | 6952±156 | 7937±167 | 52739±198 | 975±23 | 983±24 | 1288±28 | 1302±29 | **510±20** |
| | | | | | *meta-llama / LLaMA-2-7B-Chat* | | | | | |
| OP (%) ↑ | 38.94±0.3 | 34.30±1.2 | 42.09±1.6 | 40.08±1.6 | 42.36±1.2 | 36.23±0.8 | 37.69±1.2 | 41.60±0.8 | 42.78±0.8 | **43.52±1.0** |
| BWT (%) ↓ | — | 18.50±0.8 | 8.06±1.2 | 6.77±1.2 | 5.97±0.8 | 8.25±0.8 | 8.03±0.8 | 7.12±0.4 | 7.16±0.4 | 3.46±0.4 |
| TIME (s) ↓ | — | 1132±26 | 6416±145 | 7385±158 | 50283±195 | 899±22 | 912±23 | 1286±27 | 1293±28 | **485±18** |
| | | | | | *google / Gemma-2B-it* | | | | | |
| OP (%) ↑ | 32.30±0.2 | 31.89±0.8 | 32.85±1.4 | 26.48±1.5 | 28.35±1.6 | 31.14±1.2 | 32.42±1.0 | 33.25±0.9 | **33.73±0.8** | 33.41±0.9 |
| BWT (%) ↓ | — | 15.28±0.4 | 12.27±0.9 | 18.25±0.9 | 16.96±1.2 | 15.77±0.7 | 14.25±0.5 | 13.66±0.5 | 12.36±0.4 | 8.50±0.5 |
| TIME (s) ↓ | — | 510±15 | 2534±85 | 2892±92 | 17791±185 | 413±13 | 415±13 | 598±16 | 592±16 | **271±15** |
| | | | | | *meta-llama / LLaMA-3.2-1B-Instruct* | | | | | |
| OP (%) ↑ | 31.16±0.4 | 29.73±1.6 | 30.12±2.0 | 32.19±2.0 | 31.96±1.6 | 29.38±1.2 | 30.76±1.2 | 33.73±1.2 | 32.94±0.8 | **36.14±0.7** |
| BWT (%) ↓ | — | 17.03±1.2 | 15.20±1.6 | 10.74±1.6 | 11.62±1.2 | 13.57±0.8 | 11.34±0.8 | 12.36±0.8 | 12.89±1.2 | 7.36±0.8 |
| TIME (s) ↓ | — | 482±14 | 1345±45 | 1547±52 | 8893±165 | 270±11 | 281±11 | 450±14 | 448±14 | **179±13** |

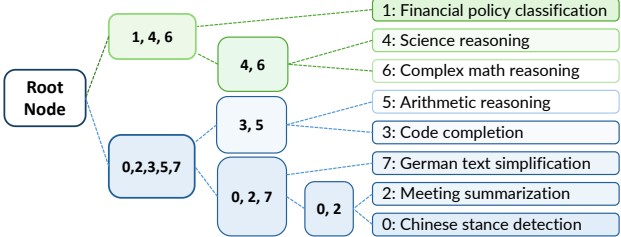

**Figure 5:** Visualization of the tree structure learned by our approach, using Mistral-7B-Instruct-v0.3 as the foundation model.

This is only marginally lower than the full 20-epoch training (88.54%), while requiring only 11% of the training time. The rapid convergence can be attributed to our bandit-based search algorithm, which efficiently explores the tree structure, as well as to TreeLoRA's ability to facilitate knowledge transfer through its hierarchical design, allowing new tasks to leverage shared knowledge from previously learned ones.

## 5.2. Evaluation on Large Language Models

In this part, we evaluate the performance of our TreeLoRA on the *large language models* (LLMs).

**Dataset and Contenders.** In this part, we employ the TRACE dataset (Wang et al., 2023c), a benchmark designed especially for continual learning with LLMs. It consists of 8 distinct sub-datasets with 200, 000 samples, containing challenging tasks including multilingual capabilities, code generation, and mathematical reasoning.

**Model Architectures.** We evaluate all methods on four different foundation LLMs: Mistral-7B-Instruct-v0.3, LLaMA-2-7B-chat, Gemma-2B-it, and LLaMA-3.2-1B-

**Table 4:** Impact of the LoRA depth (aka tree depth in TreeLoRA) on the performance for O-LoRA and TreeLoRA, where we use *meta-llama / LLaMA-2-7B-Chat* as the foundation model.

| Depth | O-LoRA | TreeLoRA |
|---|---|---|
| 8 | 21.43 ± 0.6 | 21.49 ± 0.6 |
| 16 | 22.05 ± 0.8 | 22.62 ± 0.9 |
| 32 | 37.51 ± 0.7 | 38.62 ± 0.8 |
| 64 | 42.78 ± 0.8 | 43.52 ± 1.0 |

Instruct. These models represent a diverse range of architectures and parameter scales, allowing to assess efficiency and effectiveness across different model sizes and architectures.

**Results Analysis of LLMs Experiments.** As shown in Table 3, TreeLoRA consistently achieves the best performance, outperforming other baselines and surpassing previous state-of-the-art methods, *HiDeLoRA* and *O-LoRA*. This superior performance can be attributed to TreeLoRA's hierarchical task structure, which effectively captures and leverages shared knowledge between similar tasks, while separating conflicting tasks into different branches to alleviate catastrophic forgetting. Notably, TreeLoRA also demonstrates superior computational efficiency, reducing training time by approximately 2.4× compared to other advanced methods across all model sizes.

**Visualization of the Learned Tree Structure.** To answer question **Q3**, we visualize the hierarchical task structure learned by our approach. Figure 5 illustrates the hierarchical task structure learned by TreeLoRA, using Mistral-7B-Instruct-v0.3 as the foundation model. The tree structure effectively captures the semantic relationships between different tasks. For example, tasks with similar linguistic characteristics are grouped together: C-STANCE (Chinese

**Table 5:** Ablation study of our approach. `Penalty` indicates the penalty term in Eq. (3). `LCB-Search` indicates our bandit search strategy based on lower confidence bound (LCB) for selecting the best task group as in Section 3.2.

| ID | Penalty | LCB-Search | OP (%) |
|---|---|---|---|
| (i) | - | - | $51.92 \pm 1.3$ |
| (ii) | $\checkmark$ | - | $54.03 \pm 1.2$ |
| *TreeLoRA* | $\checkmark$ | $\checkmark$ | **$54.77 \pm 1.1$** |

**Table 6:** Hyperparameter sensitivity analysis on Split CIFAR-100. We report accuracy (%) with a standard deviation over three runs.

| Reg coeff $\lambda$ | | Learning rate $\alpha$ | |
|---|---|---|---|
| 0.1 | $88.54 \pm 0.05$ | 0.001 | $87.97 \pm 0.05$ |
| 0.3 | $88.58 \pm 0.04$ | 0.003 | $88.56 \pm 0.15$ |
| 0.5 | $88.51 \pm 0.03$ | 0.005 | $88.54 \pm 0.05$ |
| 0.7 | $88.49 \pm 0.05$ | 0.007 | $88.56 \pm 0.03$ |
| 1.0 | $88.52 \pm 0.02$ | 0.010 | $88.34 \pm 0.17$ |

stance detection) and MeetingBank (meeting summarization) are placed under the same parent node, as they both involve understanding and analyzing natural language discourse. Similarly, tasks requiring mathematical and logical reasoning capabilities, such as ScienceQA (science reasoning) and NumGLUE-ds (complex math problems), are clustered nearby. This answers the research question **Q3** that the proposed TreeLoRA is able to capture the inherent task similarity structure effectively on the fly.

**Impact of Tree Depth.** To further explore the impact of tree depth, we conducted experiments to validate how varying tree depth affects the model's performance and training efficiency. The results for LLM settings are reported in Table 4, which show that TreeLoRA is relatively robust to the choice of LoRA depth. TreeLoRA consistently outperforms O-LoRA across different LoRA depths, and a depth of 64 is recommended for the best trade-off between performance and efficiency. It is important to note that LoRA depth is not determined by the number of tasks, as a single node in the tree can contain multiple tasks, allowing it to scale to a large number of tasks. Additionally, the maximum tree depth should not exceed the number of transformer layers.

**Ablation Study.** To validate each component's contribution in TreeLoRA, we evaluate it on *mistralai / Mistral-7B-Instruct-v0.3* with three variants: (i) a baseline model without the gradient-similarity penalty term or LCB-based searching, (ii) a model with only the penalty term but using a random search strategy, and (iii) our complete TreeLoRA approach with both the gradient-similarity penalty term and LCB-based search strategy. As shown in Table 5, both components contribute substantially to the overall performance. The penalty term improves performance by 2.11% (from 51.92% to 54.03%), demonstrating its effectiveness in guiding sparse parameter updates. Adding the LCB-based search strategy further enhances performance, showing the value of bandit search strategy. When combined, these components work together to achieve the best performance, demonstrating that both are crucial for TreeLoRA's effectiveness.

**Hyperparameter Sensitivity Analysis.** We conduct hyperparameter sensitivity analysis to evaluate the robustness of TreeLoRA. Our analysis focuses on two key hyperparameters: regularization coefficient $\lambda$, and learning rate $\alpha$. For $\lambda$

analysis, we fix the learning rate at default value $\alpha = 0.005$ but vary $\lambda$ from 0.1 to 1.0; while for learning rate analysis, we fix $\lambda = 0.1$ but vary $\alpha$ from 0.001 to 0.010. Table 6 summarizes our findings across different hyperparameter settings. Note that in our proposed TreeLoRA, the hyperparameter of task-similarity threshold $\delta$ in Eq. (2) does not need to be tuned, as it is automatically determined when we construct the K-D tree at the end of each task.

The regularization coefficient ($\lambda$) shows stability in model performance, with accuracy consistently around 88.50% across a wide range of values (0.1 to 1.0). This demonstrates TreeLoRA's inherent robustness to this hyperparameter. Similarly, the learning rate ($\alpha$) exhibits stable performance within the range $[0.003, 0.007]$, with peak accuracy of 88.56% at both $\alpha = 0.003$ and $\alpha = 0.007$. While extremely small or large learning rates can affect the performance, TreeLoRA maintains consistent performance across a broad range of settings. Based on these experiments, we recommend default values of $\lambda = 0.1$ and $\alpha = 0.005$.

Due to page limits, we defer more experiments including extended datasets, comparators, and models to Appendix A.

## 6. Conclusion

In this paper, we investigate the problem of efficient continual learning, especially the approaches that are tailored to LPMs. We introduce TreeLoRA, a novel approach that constructs *layer-wise* adapters by leveraging hierarchical gradient similarity to facilitate efficient knowledge sharing across tasks. To deal with the computational challenge in task similarity estimation, we develop a bandit-based algorithm utilizing the lower confidence bound for efficient tree structure exploration, along with sparse gradient updates to better suit the LPMs. Theoretically, our approach enjoys tighter regret bounds compared to conventional bandit methods that do not employ the tree structure. Experiments on both vision transformers and large language models validate the effectiveness and efficiency of our approach, with up to $3.2\times$ speedup for ViTs and $2.4\times$ speedup for LLMs, while maintaining or even improving the performance. In the future, we plan to extend our framework to consider the more challenging problem of evolving task relationships, where the task structure is time-varying.

## Acknowledgements

This research was supported by National Science Foundation of China (U23A20382, 62176117) and Postgraduate Research & Practice Innovation Program of Jiangsu Province (KYCX25_0323). Zhen-Yu Zhang was supported by JSPS KAKENHI Grant Number JP25K21282.

## Impact Statement

This paper proposes an efficient continual learning method that could significantly reduce computational resources for fine-tuning, which improves efficiency and sustainability in AI development. By reducing computational demands, our method may help to decrease energy consumption and carbon emissions associated with training AI models, contributing to environmentally sustainable machine learning.

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

# A. Additional Experiments

This section provides additional experimental details for the main paper.

## A.1. Details of Datasets

For the experiments on ViTs, we evaluate on the following three widely-used benchmark datasets:

- *Split CIFAR-100* (Zenke et al., 2017): A widely-used continual learning benchmark that divides CIFAR-100 into 10 sequential tasks, each containing 10 disjoint classes. CIFAR-100 consists of 100 classes, with each class containing 600 numbers of $32 \times 32$ color images (500 for training and 100 for testing). Each task comprises 5, 000 training samples and 1, 000 test samples.

- *Split ImageNet-R* (Wang et al., 2022a): A challenging benchmark that segments ImageNet-R's 200 classes into 10 distinct tasks, with each task containing 20 classes. The original ImageNet-R dataset comprises 30, 000 images (24, 000 for training and 6, 000 for testing), consisting of artistic renditions (including cartoons, graffiti, paintings, origami, and other artistic styles) and challenging variants of ImageNet samples.

- *Split CUB-200*: A fine-grained benchmark based on CUB-200-2011 (Wah et al., 2011), which contains 200 different bird species divided into 10 sequential tasks (20 classes per task). The original CUB-200-2011 dataset comprises 11, 788 images (5, 994 for training, 5, 794 for testing) across 200 bird species.

For the experiments on LLMs, we evaluate on the following three widely-used benchmark datasets:

- *TRACE* (Wang et al., 2023c): A benchmark designed especially for the continual learning with LLMs. It consists of 8 distinct datasets spanning challenging tasks, including domain-specific tasks, multilingual capabilities, code generation, and mathematical reasoning, with each task contains 5, 000 instances. Specifically, the eight tasks are: C-STANCE, FOMC, MeetingBank, Py150, ScienceQA, NumGLUE-cm, NumGLUE-ds, and, 20Minuten. All datasets are standardized into a unified format, allowing for effortless automatic evaluation of LLMs. Therefore, it contains a total of 200, 000 samples, with 40, 000 training examples and 16, 000 testing examples.

Note that TRACE contains a wide range of different tasks, including domain-specific tasks, multilingual capabilities, code generation, and mathematical reasoning. The performance measure for each task is different: For C-STANCE and FOMC tasks, we use accuracy as the evaluation metric to assess the model's classification performance. MeetingBank task performance is evaluated using the ROUGE-L score, which measures the longest common subsequence between the generated and reference summaries. For code-related task Py150, we employ a similarity score to evaluate the quality of generated code. The ScienceQA task is evaluated using accuracy to measure the correctness of scientific question answering. Both NumGLUE-cm and NumGLUE-ds tasks, which focus on mathematical reasoning, use accuracy as their evaluation metrics. Lastly, for the multilingual task 20Minuten, we utilize the SARI score to assess the quality of text simplification.

## A.2. Details of Contenders

In our experiments, we compare TreeLoRA with the following baselines and state-of-the-art continual learning methods:

- *SeqLoRA*: A baseline method that sequentially learns new LoRA adapters for the data stream when a new task arrives.

- *GEM* (Lopez-Paz & Ranzato, 2017): A rehearsal-based CL method that stores a small set of samples from each task to record their gradients, and performs gradient projection when the new task's gradient conflicts with previous task ones.

- *EWC* (Kirkpatrick et al., 2017): A regularization-based continual learning method that uses the Fisher Information Matrix to identify and protect parameters that are important for previous tasks.

- *L2P* (Wang et al., 2022b): A prompt-based continual learning method that learns a prompt pool shared by all tasks, where each prompt is associated with a key vector, and instructs the last multi-self attention (MSA) layer in a Prompt Tuning fashion.

- *DualPrompt* (Wang et al., 2022a): Extends L2P by introducing both general and task-specific expert prompts to better handle knowledge transfer, and instruct different MSA layers in a Prefix-Tuning fashion.

- *HiDePrompt* (Wang et al., 2023a): A hierarchical decomposition prompt learning method that decomposes continual learning objective into hierarchical components: within-task prediction, task-identity inference, and task-adaptive

prediction. It uses task-specific prompts to replace all task-sharing prompts, and then replaces the task-specific keys with a task-identity inference output layer.

- *HiDeLoRA* (Wang et al., 2024a): A generalized extension of HiDePrompt that replaces prompt-based parameter-efficient tuning (PET) with a unified framework for CL with pre-trained models and PET. A hierarchical decomposition approach for LoRA adapters to enable efficient parameter sharing.

- *FIX (ICL)* (Brown et al., 2020): An in-context learning method that incorporates 6-shot task demonstrations into the prompt, serving as a baseline to demonstrate the model's performance without continual learning.

- *O-LoRA* (Wang et al., 2023b): A method that learns tasks in different orthogonal low-rank subspaces to minimize interference between tasks.

- *SAPT* (Zhao et al., 2024b): A continual learning method that aligns parameter-efficient tuning with task selection through a shared attention mechanism.

- *TASL* (Feng et al., 2024): A continual learning method based on skill localization that identifies a small subset of parameters responsible for task-specific performance.

- *InfLoRA* (Liang & Li, 2024): A parameter-efficient fine-tuning method that injects a small number of parameters to reparameterize the pre-trained weights within a subspace to diminish interference between new and old tasks.

### A.3. Implementation Details

**Implementation Details of ViTs.** For the ViTs experiments, we follow similar implementation details to prior works (Wang et al., 2023a; Smith et al., 2023). We adopt a pre-trained ViT-B/16 model iBOT-21K, a self-supervised model pre-trained on ImageNet-21K that achieves state-of-the-art classification performance as the backbone network. The model is optimized using Adam with $\beta_1 = 0.9$ and $\beta_2 = 0.999$, utilizing a batch size of 192 and a constant learning rate at 0.005. We train 20 epochs on Split CIFAR-100 and 50 epochs on other benchmarks to ensure convergence. Input images are preprocessed to $224 \times 224$ dimensions and normalized to the range $[0, 1]$. We conduct the baselines following their original implementations. Specifically, *L2P* (Wang et al., 2022b) is implemented with a prompt pool size set as 30, and prompt length as 5. *DualPrompt* (Wang et al., 2022a) employs length-5 task-shared prompts in layers 1-2 and length-20 task-specific prompts in layers 3-5. *GEM* (Lopez-Paz & Ranzato, 2017) is implemented with a memory size of 100 samples and memory strength of 0.05. *EWC* (Kirkpatrick et al., 2017) is implemented with a regularization strength of 50. We set hyperparameters for *HiDePrompt* following Wang et al. (2023a). *HiDeLoRA* (Wang et al., 2024a) is implemented with a prompt length as 20, and LoRA low-dimension bottleneck as 10. Note that our approach can be seamlessly integrated into the existing CL frameworks. In our implementation, we incorporate HiDeLoRA's task decomposition methodology into TreeLoRA. The reported accuracy is calculated as the average test accuracy across all tasks after training on the final task. All experiments are conducted using eight Nvidia V100 GPUs with two Intel(R) Xeon(R) Gold 6248R CPUs.

**Implementation Details of ViTs and LLMs.** For the LLMs experiments, we follow the same learning rate and training epochs following the original setting of the TRACE dataset (Wang et al., 2023c), i.e., $\alpha = 1e - 5$ and epochs=1, 1, 5, 5, 1, 5, 5, 5 for the methods without LoRA adapters (*OGD*, *GEM*, and *EWC*); and $\alpha = 1e - 4$ with epochs=5, 3, 7, 5, 3, 5, 5, 7 for the methods with LoRA adapters (*SeqLoRA*, *L2P*, *DualPrompt*, *HiDeLoRA*, and *O-LoRA*). Our TreeLoRA employs a smaller number of training epochs as 2, 1, 3, 2, 1, 2, 2, 3 owing to its ability to capture the task similarity structure, while other methods using the same reduced number of epochs lead to a performance drop. For example, *O-LoRA* suffers a performance drop from 33.73% to 30.60% on the TRACE dataset with *Gemma-2b-it* when using TreeLoRA's epoch configuration.

To further control the storage complexity, TreeLoRA employs a merging strategy in which leaf nodes with highly similar gradient directions are combined when total storage exceeds a predefined threshold, integrating their LoRA adapters into a single one. The merging threshold is dynamically adjusted based on available system resources.

The batch size is set to 4 for all methods. We set the $\lambda$ as 0.5 for the LLMs experiments. All experiments in this section were performed using four Nvidia A800 (80 GB) GPUs with two Intel(R) Xeon(R) Gold 6430 CPUs. The learning rate is set to $1 \times 10^{-3}$ for all methods. We use DeepSpeed with ZeroStage 2 and mixed-precision training (BF16). The max prompt length is set to 1024 tokens.

We evaluate all the methods with four foundation large language models. These models represent a diverse range of architectures and parameter scales, allowing us to assess TreeLoRA's efficiency and effectiveness across different model sizes and architectures. We describe the key characteristics of each model below:

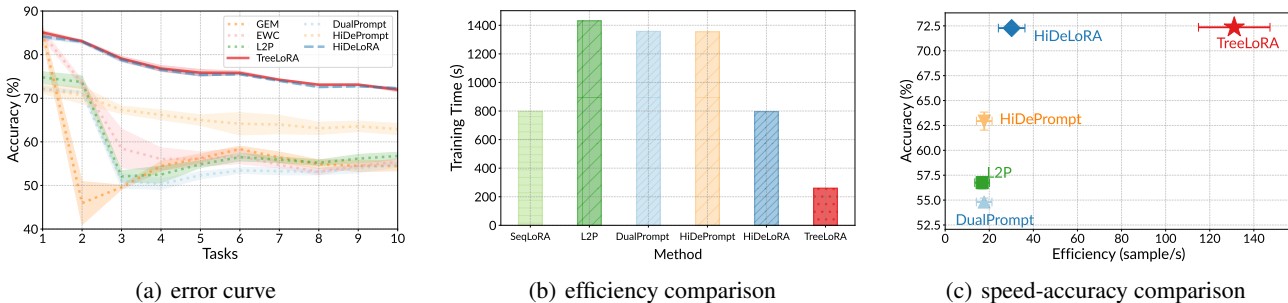

(a) error curve      (b) efficiency comparison      (c) speed-accuracy comparison

**Figure 6:** (a) Comparison of performance on the Split ImageNet-R. (b) Comparison of the efficiency of different methods. (c) Comparison of accuracy and efficiency (with mean and standard deviation over three runs). The closer to the top-right corner, the better the algorithm.

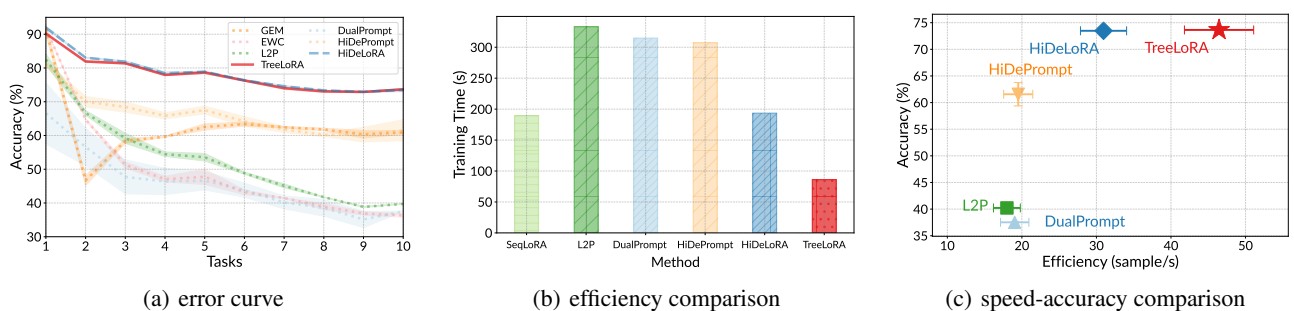

(a) error curve      (b) efficiency comparison      (c) speed-accuracy comparison

**Figure 7:** (a) Comparison of performance on the Split CUB-200. (b) Comparison of the efficiency of different methods. (c) Comparison of accuracy and efficiency (with mean and standard deviation over three runs). The closer to the top-right corner, the better the algorithm.

- *Mistral-7B-Instruct-v0.3*: An advanced instruction-tuned model utilizing sliding window attention and grouped-query attention mechanisms. The architecture comprises 32 transformer layers with 4096-dimensional hidden states, supporting a 32k token context window through rotary positional embeddings.

- *LLaMA-2-7B-chat*: A decoder-only transformer architecture with 7 billion parameters, trained on 2 trillion tokens. The model incorporates supervised fine-tuning (SFT) and constitutional AI alignment techniques, featuring 32 transformer layers with 4096-dimensional hidden states and 32 attention heads.

- *Gemma-2B-it*: A lightweight, state-of-the-art open models from Google, containing 2 billion parameters. The model employs multi-query attention mechanisms and features 18 transformer layers. It incorporates responsible AI constraints and demonstrates strong performance on reasoning and instruction-following tasks.

- *LLaMA-3.2-1B-Instruct*: A compact instruction-tuned language model comprising 1 billion parameters, optimized for efficient deployment. The architecture consists 2048-dimensional hidden states and 16 attention heads, trained on a curated subset of instruction-following data.

**Disk Storage and GPU Memory Usage.** In all experiments, the storage and GPU memory requirements of our TreeLoRA remain consistent and minimal, only requiring approximately 8 GB of GPU memory with 1 MB of additional disk storage for ViTs, 40 GB of GPU memory with 10 MB of additional disk storage for 7B LLMs, and 22 GB of GPU memory with 5 MB of additional disk storage for 2B LLMs during the entire training process. These requirements are almost the same as those of the baseline *SeqLoRA* method, ensuring that our TreeLoRA does not impose any additional storage or GPU memory burden compared compared to the vanilla LoRA.

### A.4. Additional Results on ViTs

Figure 6 and Figure 7 provide additional experimental results on ViTs. On Split ImageNet-R, as shown in Figure 6(a), TreeLoRA achieves competitive error rates, performing slightly worse than HiDeLoRA but outperforming other baseline

**Table 7:** Comparison of different continual learning methods on various foundation models on the our collected Math-LLM dataset. For each method, we report the overall performance (OP (%) ↑), and the backward transfer (BWT (%) ↓) metrics, as well as training time. Results are averaged over three runs with standard deviations. The best results are highlighted in bold.

|  | FIX (ICL) | SeqLoRA | OGD | GEM | EWC | L2P | DualPrompt | HiDeLoRA | O-LoRA | TreeLoRA |
|---|---|---|---|---|---|---|---|---|---|---|
|  | | | | | *Math-LLM Dataset* | | | | | |
| OP (%)↑ | 36.92±0.5 | 40.24±1.0 | 42.09±1.4 | 40.75±1.4 | 37.25±1.1 | 31.67±1.0 | 37.72±1.0 | 42.83±0.7 | 42.00±1.0 | **45.59±1.2** |
| BWT (%)↓ | — | 9.12±0.8 | 8.06±0.5 | 16.25±0.5 | 17.50±0.6 | 2.54±0.4 | 9.54±0.7 | 11.92±0.5 | 11.67±0.6 | 2.32±0.6 |

methods. The efficiency comparison in Figure 6(b) reveals that TreeLoRA maintains strong computational efficiency while achieving this competitive performance. Similar trends are observed on the Split CUB-200 dataset, where TreeLoRA demonstrates consistently higher accuracy than all other methods. Furthermore, our TreeLoRA is faster than the SeqLoRA, primarily due to L1-norm sparsification in Eq. (3), which reduces the number of parameters that need to be updated, and the hierarchical tree structure enables a more efficient statistical extraction of task relationships.

We also present results on the specially collected Math-LLM dataset, which includes three tasks related to math and reasoning: ScienceQA (Lu et al., 2022), NumGLUE-cm, and NumGLUE-ds (Mishra et al., 2022). As shown in Table 7, using *meta-llama / LLaMA-2-7B-chat* as the foundation model, TreeLoRA achieves the best performance among all methods, and shows a higher improvement than the performances on the original TRACE dataset. This is attributed to TreeLoRA's ability to capture the inherent task similarity structure, which is particularly advantageous for task streams with similar characteristics.

### A.5. Comparison of FLOPs and Parameter Complexity

To further compare the FLOPs of different methods, we include a comparison using *LLaMA-2-7B-Chat* as an example for training a single token on the 10th task, as shown in Table 8. Here, $m$ and $n$ denote the dimensions of the transformer's parameter matrix, $r$ is the LoRA rank, and $N$ is the number of tasks. While TreeLoRA introduces a tree structure to capture task relationships, the additional overhead for tree maintenance is minimal compared to the overall training cost.

**Table 8:** Comparison of FLOPs and parameter complexities of different methods, where we instantiate the FLOPs using *meta-llama / LLaMA-2-7B-Chat* as the foundation model, where we need to train a single token on the 10th task.

| Method | FLOPs | Parameter Complexity |
|---|---|---|
| OGD | $2.8 \times 10^{10}$ | $\mathcal{O}(mn)$ |
| LoRA | $4.2 \times 10^{6}$ | $\mathcal{O}((m+n)r)$ |
| O-LoRA | $4.2 \times 10^{7}$ | $\mathcal{O}((m+n)rN)$ |
| TreeLoRA | $4.2 \times 10^{6}$ | $\mathcal{O}((m+n)r + Nr)$ |

### A.6. Validation on More Datasets and Task Orders

To validate the stability and robustness of TreeLoRA over long task sequences, we conduct additional experiments on a sequence of 15 diverse tasks, including C-STANCE, FOMC, MeetingBank, Py150, ScienceQA, NumGLUE-cm, NumGLUE-ds, 20Minuten, dbpedia, amazon, yahoo, agnews, yelp, BoolQA, and QQP, using *meta-llama / Llama-3.2-1B-Instruct* as the foundation model. The results are summarized in Table 9. The results demonstrate that TreeLoRA maintains stable performance even with a long sequence of 15 diverse tasks, achieving higher average accuracy and lower forgetting compared to other contenders. Moreover, TreeLoRA shows even better efficiency than short-term task sequences, indicating its scalability for long-term continual learning scenarios.

**Table 9:** Performance comparison on a long sequence of 15 tasks for LLMs, where we use *meta-llama / Llama-3.2-1B-Instruct* as the foundation model.

|  | FIX | SeqLoRA | OGD | GEM | EWC | L2P | DualPrompt | HiDeLoRA | O-LoRA | TreeLoRA |
|---|---|---|---|---|---|---|---|---|---|---|
| OP (%)↑ | 41.32 | 40.71 | 32.52 | 35.48 | 31.46 | 41.05 | 41.29 | 42.38 | 44.02 | **45.68** |
| BWT (%)↓ | — | 15.72 | 21.32 | 18.33 | 22.22 | 14.92 | 15.58 | 11.23 | 10.99 | **6.41** |
| TIME (s)↓ | — | 721 | 1921 | 2235 | 13058 | 403 | 411 | 683 | 679 | **251** |

We conduct additional experiments to validate TreeLoRA and other contenders using the same datasets (i.e., MNLI, CB, WiC, COPA, QQP, BoolQA, RTE, IMDB, Yelp, Amazon, SST-2, DBpedia, Agnews, MultiRC, and Yahoo) and the same task order settings as in O-LoRA (Wang et al., 2023b). We use *meta-llama / Llama-3.2-1B-Instruct* as the foundation model. The results are shown in Table 10. The results indicate that TreeLoRA achieves stable performances across different task orders, and also improves efficiency (about 1.5x speedup compared to O-LoRA).

**Table 10:** Comparison of overall performance (%) / BWT (%) and training time (s) for different task orders on the same datasets and the same task orders as in O-LoRA (Wang et al., 2023b).

| Task Order | FIX | OGD | GEM | SeqLoRA | HiDeLoRA | O-LoRA | TreeLoRA | Task Order | FIX | HideLoRA | O-LoRA | TreeLoRA |
|---|---|---|---|---|---|---|---|---|---|---|---|---|
| Order1 | 48.75 / – | 54.16 / 10.82 | 54.10 / 10.25 | 55.71 / 8.63 | 56.32 / 2.75 | 59.50 / 2.51 | **59.73 / 2.22** | Order4 | 52.13 / – | 59.44 / 4.33 | 59.89 / 4.67 | 58.45 / 4.98 |
| Order2 | 48.75 / – | 46.82 / 21.50 | 46.70 / 20.93 | 45.26 / 7.70 | 53.41 / 5.58 | 52.53 / 5.65 | **53.78 / 5.74** | Order5 | 52.13 / – | 54.49 / 7.52 | 57.05 / 4.42 | 58.12 / 3.31 |
| Order3 | 48.75 / – | 43.79 / 27.31 | 51.12 / 16.24 | 49.03 / 19.05 | 61.25 / 3.12 | 63.82 / 2.03 | 62.76 / 2.23 | Order6 | 52.13 / – | 57.26 / 6.98 | 58.02 / 4.73 | 59.00 / 4.12 |
| Time | — | 684 | 712 | 43 | 56 | 58 | **45** | Time | — | 124 | 121 | **83** |

## A.7. Comparison with More Contenders

We conduct extended experiments to directly compare the performance of TreeLoRA and InfLoRA (Liang & Li, 2024) on the CIFAR-100 dataset using ViT models. The results are shown in Table 11. The results demonstrate that TreeLoRA achieves similar or better accuracy compared to InfLoRA, with lower training times.

**Table 11:** Comparison of TreeLoRA and InfLoRA on CIFAR-100 with ViT models.

|  | InfLoRA | TreeLoRA |
|---|---|---|
| ACC (%) ↑ | 85.44 | **88.54** |
| BWT (%) ↓ | 4.82 | **4.37** |
| TIME (s) ↓ | 695 | **214** |

Besides, we further add a comparison with two recent advanced continual learning methods, SAPT (Zhao et al., 2024b) and TASL (Feng et al., 2024), as well as other baselines, using *meta / LLaMA-2-7B-Chat* as the foundation model. For a fair comparison, generative replay is not employed in SAPT as it may introduce additional information beyond the original data stream. The results are summarized in Table 12. As it illustrated in the table, TreeLoRA achieves the best overall performance in terms of accuracy, forgetting, and efficiency compared to all baselines, including SAPT and TASL.

**Table 12:** Performance comparison with more contenders including SAPT and TASL on LLMs.

|  | FIX | SeqLoRA | OGD | GEM | EWC | L2P | DualPrompt | HiDeLoRA | O-LoRA | SAPT | TASL | TreeLoRA |
|---|---|---|---|---|---|---|---|---|---|---|---|---|
| OP (%) ↑ | 38.94 | 34.30 | 42.09 | 40.08 | 42.36 | 36.23 | 37.69 | 41.60 | 42.78 | 42.93 | 43.19 | **43.52** |
| BWT (%) ↓ | — | 18.50 | 8.06 | 6.77 | 5.97 | 8.25 | 8.03 | 7.12 | 7.16 | 5.49 | 4.58 | **3.46** |
| TIME (s) ↓ | — | 1132 | 6416 | 7385 | 50283 | 899 | 912 | 1286 | 1293 | 1205 | 1185 | **485** |

## A.8. Experiments on Extreme-Scale LLMs (13B Model)

To further investigate the computational trade-offs for extreme-scale LLMs, we conduct experiments using a 13B model (*meta / Llama-2-13b-chat-hf*). The results are reported in Table 13. The results show that TreeLoRA achieves better accuracy while using lower training time compared to other contenders. Additionally, the memory (storage) overhead of TreeLoRA is minimal, requiring only 15 MB. These findings demonstrate the effectiveness of our tree-based adaptation strategy in both performance and efficiency aspects, and its scalability to large-scale models.

**Table 13:** Performance comparison on a 13B LLM (*meta / Llama-2-13b-chat-hf*).

|  | FIX | SeqLoRA | OGD | GEM | EWC | HiDeLoRA | O-LoRA | TreeLoRA |
|---|---|---|---|---|---|---|---|---|
| OP (%) ↑ | 40.15 | 39.16 | 42.32 | 43.77 | 41.23 | 43.27 | 44.32 | **47.13** |
| BWT (%) ↓ | — | 15.58 | 9.72 | 8.42 | 10.12 | 11.27 | 5.19 | **3.42** |
| TIME (s) ↓ | — | 1525 | 8712 | 9931 | 67819 | 1835 | 1839 | **662** |

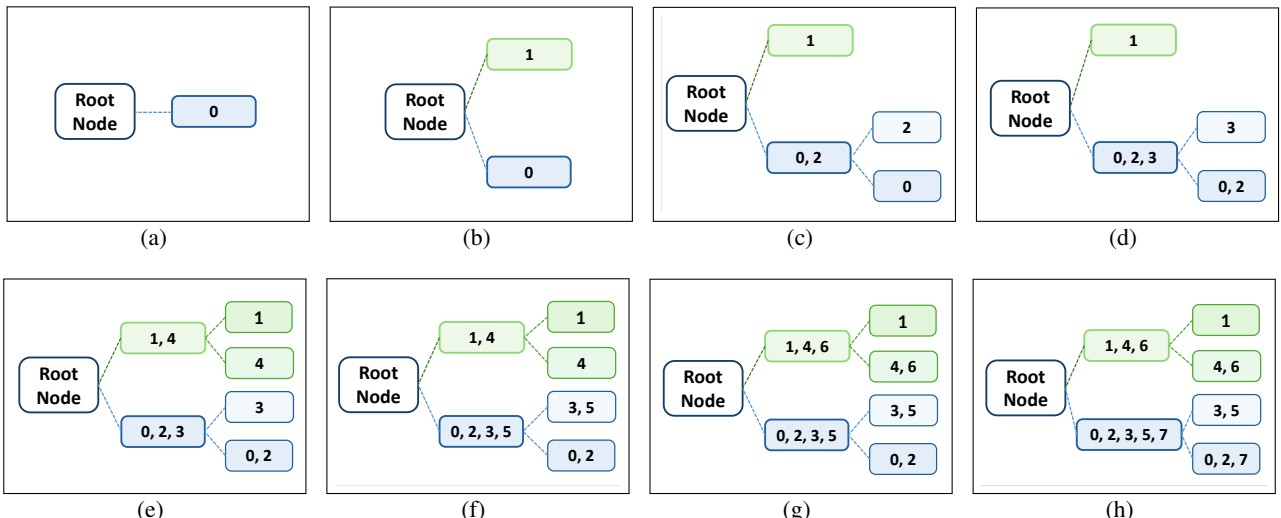

**Figure 8:** The evolution of the tree structure in our *TreeLoRA* under a dynamic task flow in our LLM experiment, using *mistralai / Mistral-7B-Instruct-v0.3* as the foundation model. Figures (a) to (h) correspond to task 1 to task 8 (i.e., $t = 1$ to $t = 8$), respectively. These figures demonstrate how the tree structure evolves over time and effectively captures the relationships among different tasks.

## A.9. Evolving of the Tree Structure

We also provide the evolution of the tree structure in our *TreeLoRA* in our LLM experiment, using *mistralai / Mistral-7B-Instruct-v0.3* as foundation model. As shown in Figure 8, (a) to (h) correspond to task 1 to task 8 (i.e., $t = 1$ to $t = 8$), respectively. These demonstrate how the tree structure evolves over time and effectively captures the task relationships.

# B. Related Work

This section introduces related works, including continual learning, low-rank approximation, and mixture of experts.

**Continual Learning.** The goal of continual learning is to continuously learn from new data while retaining knowledge of previously learned tasks (Dohare et al., 2024; Wang et al., 2024b; Zhou et al., 2024). A major challenge is catastrophic forgetting (French, 1993; McCloskey & Cohen, 1989; Thrun & Pratt, 1997), where the model forgets previously learned tasks when fine-tuned on new data. There are mainly three main approaches to deal with this issue:

Regularization-based methods (Schwarz et al., 2018) place constraints on the model parameters to ensure new tasks are learned without forgetting the old ones. For example, elastic weight consolidation (EWC) (Kirkpatrick et al., 2017) leverages the Fisher Information Matrix to identify and protect parameters critical for previous tasks. Huszár (2018) further improves EWC by maintaining penalties only for the immediately preceding task, reducing over-emphasis on early tasks and lowering storage overhead. Memory-aware synapses (Aljundi et al., 2018) improve upon these methods by estimating parameter importance through output function sensitivity, addressing the limitation of near-zero gradients at loss function local minima.

Rehearsal-based methods (Scialom et al., 2022; Bailey, 2024) store data from previous tasks in a buffer and use it to train the model alongside the current task. Notable approaches include iCaRL (Rebuffi et al., 2017), which maintains exemplar sets of feature vectors for each class, performs nearest-neighbor classification, and selects representative samples through herding while preserving past knowledge via distillation loss. Gradient episodic memory (GEM) (Lopez-Paz & Ranzato, 2017) maintains gradient information from previous tasks and employs gradient projections to resolve conflicts between current and past task gradients. Smith et al. (2024) propose an adaptive memory replay method that treats sampling of past data as a multi-armed bandit problem. While these methods have shown superior performance across various benchmarks, they typically come with a high computational and storage cost, due to the need to maintain and update the model based on a large-sized buffer of past data, which may scale linearly with the number of tasks.

Architecture-based methods (Mallya & Lazebnik, 2018; Serrà et al., 2018) dynamically adjust the model architecture to accommodate new tasks. Inspired by the complementary learning systems theory, DualNet (Pham et al., 2021) introduces a learning framework that incorporates fast and slow learning components, but this approach still relies on a rehearsal buffer for optimal performance. To address these limitations, L2P (Wang et al., 2022b) proposes a shared prompt pool strategy

that eliminates the need for explicit task identities during inference while reducing parameter and memory overhead. In this approach, each prompt is linked to a key vector and modulates the final multi-self attention (MSA) layer using Prompt Tuning techniques. Further advancing this line of research, DualPrompt (Wang et al., 2022a) enhances the L2P framework by incorporating both general-purpose and task-specific expert prompts, aimed at improving knowledge transfer across tasks. Our approach further refines the task architecture by leveraging the task similarity structure to build a *hierarchical tree structure* and *bandit-based* searching algorithm, which efficiently adapts to new tasks and achieves comparable performance.

**Supervised Fine-Tuning.** The recent advancement in large language models has demonstrated the effectiveness of supervised fine-tuning approaches. InstructGPT (Ouyang et al., 2022) shows that fine-tuning LLMs with human feedback enhances their instruction-following capabilities and response quality. To address the computational burden of fine-tuning LLMs, LoRA (Hu et al., 2022) proposes a low-rank adaptation method that substantially reduces trainable parameters while preserving model performance. While these methods have become fundamental techniques for adapting pre-trained models to downstream tasks, addressing distribution shifts between training and testing data remains challenging, particularly in continual learning scenarios where new tasks emerge continuously.

**Low-Rank Adaptation.** The low-rank approximation has emerged as an effective technique for model adaptation and compression. The pioneering work LoRA (Hu et al., 2022) introduces low-rank adaptation matrices to efficiently fine-tune large language models while preserving their capabilities. Following this direction, QLoRA (Dettmers et al., 2023) combines quantization with low-rank adaptation to further reduce memory usage. More recent works like AdaLoRA (Hu et al., 2024) propose adaptive budget allocation strategies for parameter-efficient fine-tuning, while GaLore (Lin et al., 2023) focuses on memory-efficient training through gradient low-rank projection. Our TreeLoRA also takes inspiration from these works to be better suited to the LPMs by considering the low-rank approximation.

**Mixture of Experts.** Mixture of Experts (MoE) (Yüksel et al., 2012) is a powerful architecture that allows for efficient parallel computation by distributing the workload across multiple experts. Recent works have demonstrated the effectiveness of MoE in continual learning scenarios. LoRAMoE (Dou et al., 2024) introduces a framework that combines low-rank adapters with a router network, functioning as a plugin version of MoE. By freezing the backbone model and directing specific adapters to leverage shared knowledge for downstream tasks, it effectively mitigates catastrophic forgetting. MoRAL (Yang et al., 2024) combines MoE with LoRA for continual learning of LLMs, demonstrating robust knowledge retention and superior performance through question-answer pairs rather than factual triplets. Although these approaches show promising results in continual learning, they still incur significant computational overhead due to their architectures. Instead, we propose an efficient CL approach that leverages task similarity through *hierarchical tree structure* and *bandit-based* searching algorithm, enabling fast adaptation and achieving comparable performance.

**Non-stationary Online Learning.** Non-stationary online learning aims to design algorithms capable of adapting to distribution shifts in non-stationary environments. A theoretically grounded approach is *dynamic regret minimization* (Zhang et al., 2018), which evaluates performance relative to a sequence of time-varying benchmark models. A principled framework for optimizing dynamic regret in online convex optimization is the *online ensemble* framework (Zhao et al., 2024a), which maintains a set of diverse base learners and combines them through a meta-algorithm. Recently, various algorithms have emerged from this framework (Bai et al., 2022; Qian et al., 2023; Zhang et al., 2023). For instance, ATLAS (Bai et al., 2022) addresses label shift by employing base learners with varying learning rates, while the wavelet-based method (Qian et al., 2024) detects changes and restarts using an ensemble of wavelet bases. Non-stationary online learning primarily focuses on adapting to the current task and pays less attention to the issue of catastrophic forgetting, which is more concerned with continual learning. Extending theoretical insights from non-stationary online learning could provide new perspectives for advancing continual learning theory (Evron et al., 2022).

# C. Proofs

This section presents the proofs that were omitted in the main paper.

## C.1. Proof of Theorem 1

*Proof.* We aim to analyze the regret bound for the TreeLoRA algorithm. Recall that the regret is defined as:

$$\mathbf{Reg}(T) \triangleq \mathbb{E}\left[\sum_{t=1}^{T} \hat{\xi}_t^k\right] - \min_{k^\star \in [N]} \sum_{t=1}^{T} \xi^{k^\star} \leq \sum_{j=1}^{N} \mathbb{E}[n_{j,T}] \cdot \Delta_j,$$

where $n_{j,T}$ is the number of times arm $j$ is pulled up to time $T$, and $\Delta_j \triangleq \xi^j - \xi^{k^\star}$ is the suboptimality gap for the arm $j$.

*Decomposing the Regret.* We start by decomposing the regret into two terms based on whether $\Delta_j$ is less than or greater than a threshold $\Delta_\dagger$:

$$\mathbf{Reg}(T) = \underbrace{\sum_{j \in [N]: \Delta_j \leq \Delta_\dagger} \mathbb{E}[n_{j,T}] \cdot \Delta_j}_{\texttt{term (a)}} + \underbrace{\sum_{j \in [N]: \Delta_j > \Delta_\dagger} \mathbb{E}[n_{j,T}] \cdot \Delta_j}_{\texttt{term (b)}}.$$

*Bounding* `term (a)`. Since $\Delta_j \leq \Delta_\dagger$, we bound `term (a)` as:

$$\texttt{term (a)} \leq T \cdot \Delta_\dagger.$$

*Decomposing and Bounding* `term (b)`. For `term (b)`, we further divide the summation into two parts:

$$\texttt{term (b)} = \underbrace{\sum_{j \in \mathcal{I}_{2\eta/3}} \mathbb{E}[n_{j,T}] \cdot \Delta_j}_{\texttt{term (c)}} + \underbrace{\sum_{j \notin \mathcal{I}_{2\eta/3}} \mathbb{E}[n_{j,T}] \cdot \Delta_j}_{\texttt{term (d)}},$$

where $\mathcal{I}_{2\eta/3} \triangleq [\mathrm{node}_j \in [N] : \Delta_j \leq 2\eta/3]$ is the set of nodes with suboptimality gap less than $2\eta/3$.

For `term (c)`, we consider the same event as in the proof of Theorem 3 in (Coquelin & Munos, 2007). Let $h$ be the smallest integer such that $\delta_h \leq \eta/3$. We have $h \leq \frac{\log(3\delta/\eta)}{\log(1/\gamma)} + 1$. Let $i$ be a node of depth $h$. If $i \in \mathcal{I}_{2\eta/3}$, then, thanks to Assumption 1, we have for all $j \in L(\mathcal{N}_i), \mu_j \geq \mu_i - \eta/3 \geq \mu^* - \eta$, thus $j \in J_\eta$. Therefore,

$$\texttt{term (c)} \leq \sum_{j \in J_\eta} \mathbb{E}[n_{j,T}] \cdot \Delta_j \leq \sum_{j \in J_\eta} \frac{6}{\Delta_j} \log\left(\frac{4NT}{\Delta_j^2}\right) \leq \frac{6|J_\eta|}{\Delta_\dagger} \log\left(\frac{4NT}{\Delta_\dagger^2}\right),$$

where the second inequality follows from Lemma 1.

For `term (d)`, we use the same argument as Theorem 4 in (Coquelin & Munos, 2007) and obtain that with probability at least $1 - 1/T$, we have

$$\texttt{term (d)} \leq \frac{54(3\delta)^c}{\eta^{2+c}} \log\left(\frac{4NT}{\eta^2}\right),$$

where $c \triangleq \log(2)/\log(1/\gamma)$. The term $O\left(1/\eta^{2+c}\right)$ depends weakly (linearly) on the depth $D$ (through $\log(N)$). Thus, if $\eta$ is fixed and we increase the depth $D$ of the tree, `term (d)` is *not the dominant term*.

*Combining Bounds.* Adding the bounds for `term (a)`, `term (c)`, and `term (d)`, the overall regret satisfies:

$$\mathbf{Reg}(T) \leq T \cdot \Delta_\dagger + \frac{6|J_\eta|}{\Delta_\dagger} \log\left(\frac{4NT}{\Delta_\dagger^2}\right) + \frac{54(3\delta)^c}{\eta^{2+c}} \log\left(\frac{4NT}{\eta^2}\right) \leq \mathcal{O}\left(T \cdot \Delta_\dagger + \frac{|J_\eta|}{\Delta_\dagger} \log\left(\frac{NT}{\Delta_\dagger^2}\right)\right)$$

By selecting $\Delta_\dagger$ as $\Delta_\dagger = \sqrt{|J_n|/T}$, we have:

$$\mathbf{Reg}(T) \leq \mathcal{O}\left(\sqrt{T|J_\eta| \log\left(\frac{NT}{|J_\eta|}\right)} + \frac{\delta^c}{\eta^{2+c}} \log\left(\frac{NT}{\eta^2}\right)\right),$$

which finished the proof. □

## C.2. Useful Lemmas

We present the following useful lemmas.

**Lemma 1** (Theorem 2 and Theorem 3 in (Coquelin & Munos, 2007)). *With probability at least $1 - 1/T$, for all $j \in J_\eta$, for any leaf node, the number of times a leaf $i$ is chosen is at most $n_i \leq \frac{6}{\Delta_i^2} \log\left(\frac{4 \cdot 2^D \cdot T}{\Delta_i^2}\right)$, where $D = \mathcal{O}(\log N)$ is the depth of the tree. Beside, with Assumption 1, with probability at least $1 - 1/T$, the number of times a node $i$ is visited is at most $n_i \leq \frac{6 \log\left(4NT/(\Delta_i - \delta_h)^2\right)}{(\Delta_i - \delta_h)^2}$.*

