# OpenReview forum: "TreeLoRA: Efficient Continual Learning via Layer-Wise LoRAs Guided by a Hierarchical Gradient-Similarity Tree"
_ICML.cc/2025/Conference — ICML 2025 poster_

### Official Review · Reviewer_qcDX · 2025-03-10

**Overall Recommendation:** 3

**Summary:**

This paper proposes a novel continuous learning approach, TreeLoRA (K-D Tree of Low-Rank Adapters), which exploits hierarchical gradient similarity to build layer-wise adapters for efficient CL.To achieve even greater efficiency, the authors develop a confidence lower bound based bandit techniques to efficiently explore the task structure. In addition, the authors provide theoretical analyses to demonstrate the validity of the proposed approach.

**Claims And Evidence:**

The claims made by the authors are well-supported by clear theoretical proofs and experimental results, which effectively validate their assertions.

**Essential References Not Discussed:**

N/A.

**Experimental Designs Or Analyses:**

See Weaknesses (2).

**Methods And Evaluation Criteria:**

The proposed methods are effective in addressing the problem outlined in the paper.

**Other Comments Or Suggestions:**

Typos: I'm not sure if this is a typo, but I noticed that the variable $j$ in equation (1) seems to be unnecessary, as it not be defined.

**Other Strengths And Weaknesses:**

Weaknesses:

1. The definition of $f_{i}(w_{j})$ does not conform to common notation. It is recommended to swap $i$ and $j$ to write it as $f_{j}(\mathcal{T}\_i)$, or consider using $\theta_{j}$ to represent the model parameters. This would enhance the clarity of the paper.

2. In the methods compared in this paper, it seems that there is a lack of comparison with some recent advanced continual learning methods [1, 2], which might have better performance than the approach proposed in this paper.

[1] Zhao, W., et al. Sapt: A shared attention framework for parameter-efficient continual learning of large language models. In ACL, 2024.

[2] Feng, Y., et al. Tasl: Task skill localization and consolidation for language model continual learning. In ACL, 2024.

**Questions For Authors:**

Question:

1. Regarding the use of LCB to calculate the similarity between tasks especially in transformer-based models, is it computed only at the last layer of the model, or is each layer calculated individually? I noticed that the figures in the paper seem to indicate that only the last layer is calculated, so why not compute each layer separately, given that the features learned at each layer of the model are different?

2. The authors mention that the setting of the threshold $\delta$ does not need to be done manually and is done dynamically, why and how is this done?

**Relation To Broader Scientific Literature:**

The research is a study of model base capabilities, with potential implications for the broader scientific literature.

**Theoretical Claims:**

I have checked the proof of theory provided by the author and there are no obvious problems.

---

> ### Author Rebuttal · Authors · 2025-03-31
>
> We sincerely appreciate the reviewer's constructive feedback. In the following, we respond to each question.
>
> ---
>
> **Q1.** "Regarding the use of LCB to calculate the similarity between tasks especially in transformer-based models, is it computed only at the last layer of the model, or is each layer calculated individually? I noticed that the figures in the paper seem to indicate that only the last layer is calculated, so why not compute each layer separately, given that the features learned at each layer of the model are different?"
>
> **A1.** Thank you for your question. We would like to clarify that the LCB calculation in TreeLoRA is indeed performed ***layer by layer*** across the entire model. As described in Equation (2) in our paper, the LCB is computed as follows:
>
> $$
> \mathrm{LCB}\_k=  \begin{cases}  \widehat{\mu}\_k-2 \sqrt{\frac{\log t}{n\_k}}, & \text { if } k \in \mathcal{L} \\\\
> \max \\left\\{ \min\_{j \in \mathcal{C}} \\left\\{\widehat{\mu}\_j-2 \sqrt{\frac{\log t}{n\_j}}-\delta \\right\\} \\right\\}, & \text { if } k \notin \mathcal{L}  \end{cases}
> $$
>
> where $\hat{\mu}\_k = \frac{1}{|\operatorname{Select}\_k|} \sum\_{\tau \in \{\operatorname{Select}\_k\}} \hat{\xi}\_{\tau}^k$ is the estimated task similarity between the current task and the $k$-th task group (i.e., the nodes in the branch of the selected leaf node at round $t$), $\mathcal{L}$ is the set of all leaf nodes, $\delta$ is the automatically determined threshold, and $\mathcal{C}$ is the child nodes of the $k$-th node. Therefore, the LCB is computed for each layer of the model. By calculating the LCB layer by layer, TreeLoRA captures the similarity between tasks at various levels throughout the model hierarchy as illustrated in Figure 1, allowing us to better capture hierarchical task similarities, which is especially advantageous in transformer-based models. We will add these details in the revised version of the paper to provide further clarity. Thank you again for your valuable question!
>
> ---
>
> **Q2.** "The authors mention that the setting of the threshold $\delta$ does not need to be done manually and is done dynamically, why and how is this done?"
>
> **A2.** Thanks for your comment. Inspired by the K-D tree data structure [Bentley, 1990], the threshold $\delta$ does not require manual tuning. Specifically, during the construction of the K-D tree after each task, the gradient space is partitioned based on the distribution of task gradients. At each split, the threshold is computed by taking the median of the similarity (L1-norm) between each task gradient and the mean gradient within the corresponding task group. This approach ensures balanced tree growth and adaptive partitioning of the gradient space, without the need for manual threshold adjustments.
>
> ---
>
> **Q3.** "definition of $f_i(w_j)$ does not conform to common notation. It is recommended to swap $i$ and $j$ to write it as $f_j(\mathcal{T}_i)$, or consider using $\theta_j$ to represent the model parameters"
>
> **A3.** Thanks for your comment. We will revise our paper accordingly and use clearer notations, which would enhance the clarity of the paper. Thanks again for your feedback.
>
> ---
>
> **Q4.** "it seems that there is a lack of comparison with some recent advanced continual learning methods [1, 2]"
>
> **A4.** Thank you for pointing out these two references. Following your suggestions, we add a comparison with these two recent advanced continual learning methods, SAPT [Zhao et al., ACL 2024] and TASL [Feng et al., ACL 2024]. For a fair comparison, we do not employ the generative replay in SAPT. The results, using _meta / LLaMA-2-7B-Chat_ as the foundation model, are presented in the table below:
>
> |Metric|FIX|SeqLoRA|OGD|GEM|EWC|L2P|DualPrompt|HiDeLoRA|O-LoRA|SAPT|TASL|TreeLoRA|
> |-|:-:|:-:|:-:|:-:|:-:|:-:|:-:|:-:|:-:|:-:|:-:|:-:|
> |Op (%)|38.94|34.30|42.09|40.08|42.36|36.23|37.69|41.60|42.78|42.93|43.19|**43.52**|
> |BWT (%)|-|18.50|8.06|6.77|5.97|8.25|8.03|7.12|7.16|5.49|4.58|**3.46**|
> |Time (s)|-|1132|6416|7385|50283|899|912|1286|1293|1205|1185|**485**|
>
> We also add comparison with other recent methods, InfLoRA [Liang and Li, CVPR 2024], please refer to **A3** for Reviewer bzQv for more details. We will add these results to the revised version, and will also add discussions with SAPT and TASL methods in the related work section.
>
> ---
>
> We hope these clarifications address your concerns. Thanks again for your valuable comments.

---

> > ### Comment · Reviewer_qcDX · 2025-04-03
> >
> > Thank you for providing the experiments and explanations regarding my questions and concerns. However, I still have some issues with the experimental part:
> >
> > **Q1**: You mentioned that you did not use SAPT's generative replay for a fair comparison. Why is disabling generative replay more fair? As far as I remember, the generative replay in SAPT does not use the original data but instead uses fabricated data, which should not affect fairness.
> >
> > **Q2**: In the training times you provided, O-LoRA is surprisingly close to SAPT's time. In my understanding, O-LoRA involves computing orthogonal structures for each layer and incorporating them into gradient calculations, which should be time-consuming. Or perhaps the authors considered that O-LoRA does not retain all of LoRA blocks.
> >
> > **Q3**: Why is the training time reported rather than the inference time?

---

> > > ### Author Response · Authors · 2025-04-04
> > >
> > > We are grateful to the reviewer for the follow-up feedback. We address each of the additional questions regarding experiments as follows.
> > >
> > > **Q1.** "You mentioned that you did not use SAPT's generative replay for a fair comparison. Why is disabling generative replay more fair? As far as I remember, the generative replay in SAPT does not use the original data but instead uses fabricated data, which should not affect fairness."
> > >
> > > **A1.** We thank the reviewer for the question. To clarify, the generative replay mechanism requires maintaining a pre-generated dataset of pseudo data (as observed in the SAPT's codebase) or, alternatively, employing an additional generative model to produce pseudo data. In our opinion, this process introduces additional information _beyond the original data stream_. Therefore, we exclude this mechanism and instead adopt another strategy by storing a fixed number of data samples in a buffer. This ensures that all methods rely _solely on the original data stream_.
> > >
> > > On the other hand, we also appreciate the idea of introducing generative replay in continual learning, which can be considered as a "plug-in" component. This component could be integrated into our method or O-LoRA, etc. We will conduct additional ablation studies for a more comprehensive evaluation.
> > >
> > > ---
> > >
> > > **Q2.** "In the training times you provided, O-LoRA is surprisingly close to SAPT's time. In my understanding, O-LoRA involves computing orthogonal structures for each layer and incorporating them into gradient calculations, which should be time-consuming. Or perhaps the authors considered that O-LoRA does not retain all of LoRA blocks."
> > >
> > > **A2.** We thank the reviewer for the question. We would like to clarify that although O-LoRA requires computing orthogonal structures for each layer during training, the additional computational cost remains acceptable. This is because the orthogonal regularization across different layers can be computed in a batched and parallelized manner — treating the LoRA adapters at different layers as one concatenated matrix. This strategy is implemented in both the original O-LoRA's and our codebase.
> > >
> > > ---
> > >
> > > **Q3.** "Why is the training time reported rather than the inference time?"
> > >
> > > **A3.** Thank you for your comment. In this paper, one of the key contributions of our proposed TreeLoRA is to explore the task structure in order to **facilitate adaptation to new tasks by leveraging task-shared knowledge**, therefore decreasing the training time and enhancing the efficiency. Regarding inference, our method incurs the same time cost as other LoRA-based methods since we do not modify the inference process. While reducing the inference time is also an important problem in the LLM field, our current framework is primarily designed to address the challenges associated with task adaptation speed and training overhead, and we will consider it as an important future work.
> > >
> > > ---
> > >
> > > Thanks again for your time and feedback, we hope this response addresses your concerns.

---

### Official Review · Reviewer_ygph · 2025-03-12

**Overall Recommendation:** 3

**Summary:**

TreeLoRA presents a continual learning method that enhances the efficiency of updating large pre-trained models. By integrating layer-wise LoRA with a hierarchical gradient similarity tree, it improves knowledge retention while reducing computational costs. TreeLoRA mitigates catastrophic forgetting while maintaining efficiency in VITs and LLMs.

**Claims And Evidence:**

Yes, please refer to the Strengths and Weaknesses section for more details.

**Essential References Not Discussed:**

N/A

**Ethical Review Flag:**

Flag this paper for an ethics review.

**Experimental Designs Or Analyses:**

Yes, please refer to the Strengths and Weaknesses section for more details.

**Methods And Evaluation Criteria:**

Yes, please refer to the Strengths and Weaknesses section for more details.

**Other Comments Or Suggestions:**

N/A

**Other Strengths And Weaknesses:**

***Strengths***
1. This paper presents a hierarchical gradient similarity tree for task organization, optimizing parameter updates with improved efficiency. A novel bandit-based similarity estimation reduces complexity, enhancing scalability. Sparse gradient updates further adapt TreeLoRA for ViTs and LLMs.

2. A rigorous theoretical analysis derives tighter regret bounds than standard bandit approaches. The hierarchical structure minimizes computational overhead while preserving task knowledge, ensuring efficiency gains.

3. Experiments on vision and language tasks demonstrate that TreeLoRA surpasses state-of-the-art methods, accelerates ViTs and LLMs, and mitigates catastrophic forgetting with reduced backward transfer.

***Weaknesses***
1. The paper presents evidence for TreeLoRA’s effectiveness but lacks discussion on the stability and robustness of its tree structure over extended task sequences. A deeper analysis of its evolution in long training sequences, especially in non-stationary environments, would add valuable insight.

2. Additionally, the memory and computational trade-offs for extreme-scale LLMs remain unexamined.

3. While TreeLoRA is compared to other LoRA-based continual learning methods, benchmarking against non-LoRA-based strategies is limited, such as replay-based approaches. A broader comparison would better contextualize TreeLoRA’s advantages within the continual learning landscape.

**Questions For Authors:**

N/A

**Relation To Broader Scientific Literature:**

Please refer to the Strengths and Weaknesses section for more details.

**Theoretical Claims:**

Yes, please refer to the Strengths and Weaknesses section for more details.

---

> ### Author Rebuttal · Authors · 2025-03-31
>
> Thanks for your constructive and helpful comments. We provide our response to each question as below.
>
> ---
>
> **Q1.** "The paper presents evidence for TreeLoRA's effectiveness but lacks discussion on the stability and robustness of its tree structure over extended task sequences. A deeper analysis of its evolution in long training sequences, especially in non-stationary environments, would add valuable insight."
>
> **A1.** Thank you for your comment. Following your suggestion, we conduct additional experiments to validate the stability and robustness of TreeLoRA over ***long task sequences***, which consist of a total of 15 tasks, including C-STANCE, FOMC, MeetingBank, Py150, ScienceQA, NumGLUE-cm, NumGLUE-ds, 20Minuten, dbpedia, amazon, yahoo, agnews, yelp, BoolQA, and QQP, using _meta-llama / Llama-3.2-1B-Instruct_ as the foundation model. The results are summarized in the following table:
>
> | Metric   |  FIX  | SeqLoRA |  OGD  |  GEM  |  EWC  |  L2P  | DualPrompt | HideLoRA | O-LoRA | TreeLoRA  |
> | -------- | :---: | :-----: | :---: | :---: | :---: | :---: | :--------: | :------: | :----: | :-------: |
> | Op (%)   | 41.32 |  40.71  | 32.52 | 35.48 | 31.46 | 41.05 |   41.29    |  42.38   | 44.02  | **45.68** |
> | BWT (%)  |  0.0  |  15.72  | 21.32 | 18.33 | 22.22 | 14.92 |   15.58    |  11.23   | 10.99  | **6.41**  |
> | Time (s) |   -   |   721   | 1921  | 2235  | 13058 |  403  |    411     |   683    |  679   |    **251**    |
>
> The results demonstrate that TreeLoRA maintains stable performance even with a long sequence of 15 diverse tasks, achieving higher average accuracy and lower forgetting compared to other contenders. Moreover, TreeLoRA shows even better efficiency than short-term task sequences, indicating its scalability for long-term continual learning scenarios.
>
> Additionally, we include a figure that illustrates the evolution of the tree structure under dynamic task flow in our LLM experiment. This figure helps to better visualize how TreeLoRA adapts to the evolving task structure over time. The figure is available at the following link: [https://anonymous.4open.science/r/TreeLoRA/scripts/rebuttal.jpg](https://anonymous.4open.science/r/TreeLoRA/scripts/rebuttal.jpg)
>
> ---
>
> **Q2.** "the memory and computational trade-offs for extreme-scale LLMs remain unexamined"
>
> **A2.** Thanks for your comment. In our paper, we validate our method using small ViT models, as well as large-size language models (1B, 2B, and 7B), which contain commonly used models in the research community [Wang et al., 2023a, Wang et al., 2023b, Dou et al., 2024]. To further investigate the computational trade-offs for extreme-scale LLMs, **_we add an experiment using a 13B model_** (meta / Llama-2-13b-chat-hf), as shown in the following table:
>
> | Metric   |  FIX  | SeqLoRA |  OGD  |  GEM  |  EWC  | HideLoRA | O-LoRA | TreeLoRA  |
> | -------- | :---: | :-----: | :---: | :---: | :---: | :------: | :----: | :-------: |
> | Op (%)   | 40.15 |  39.16  | 42.32 | 43.77 | 41.23 |  43.27   | 44.32  | **47.13** |
> | BWT (%)  |  0.0  |  15.58  | 9.72  | 8.42  | 10.12 |  11.27   |  5.19  | **3.42**  |
> | Time (s) |   -   |  1525   | 8712  | 9931  | 67819 |   1835   |  1839  |  **662**  |
>
> The results show that TreeLoRA achieves better accuracy while using lower training time compared to other contenders. Additionally, the memory (storage) overhead of TreeLoRA is minimal, requiring only 15 MB. These findings demonstrate the effectiveness of our tree-based adaptation strategy in both performance and efficiency aspects, and its scalability to large-scale models.
>
> ---
>
> **Q3.** "While TreeLoRA is compared to other LoRA-based continual learning methods, benchmarking against non-LoRA-based strategies is limited, such as replay-based approaches. A broader comparison would better contextualize TreeLoRA's advantages within the continual learning landscape."
>
> **A3.** Thank you for your comment. It appears there may be a misunderstanding due to our insufficient emphasis. Specifically, we have compared TreeLoRA with several non-LoRA-based continual learning strategies, including replay-based (rehearsal-based) methods, GEM, regularization-based methods such as EWC, and the baseline OGD (which represents full update, a non-LoRA method). As shown in Table 3 and Table 4 of the submission PDF, TreeLoRA outperforms these methods both in terms of performance and efficiency. Additionally, TreeLoRA offers particular advantages for transformer-based models due to the large parameter size and inherent hierarchical structure in these models.
>
> ---
>
> We hope these clarifications address your concerns. We will improve the paper writing to better emphasize these points. Thanks again for your constructive comments.

---

> > ### Comment · Reviewer_ygph · 2025-04-09
> >
> > Thanks for the author's rebuttal. After reading the comments from other reviewers, I will maintain my score.

---

> > > ### Author Response · Authors · 2025-04-09
> > >
> > > Thank you for recognizing the novelty and theoretical soundness of our work. We also greatly appreciate your insightful feedback. We will incorporate the suggested discussions and additional experiments in the revised version. Thank you again!

---

### Official Review · Reviewer_bzQv · 2025-03-13

**Overall Recommendation:** 2

**Summary:**

This paper proposes TreeLoRA, a novel and efficient approach for continual learning in large pre-trained models. TreeLoRA constructs a hierarchical tree structure of LoRAs based on gradient similarity, enabling efficient task adaptation and knowledge sharing. The method employs bandit algorithms to explore task-similarity structure and leverages sparse gradient updates to optimize parameters, demonstrating superior efficiency and performance compared to previous state-of-the-art continual learning methods.

## update after rebuttal

Thank the authors for the detailed follow-up and additional experimental results. I appreciate the authors' efforts to extend the evaluation to the full 15-task benchmark and to clarify the role of LoRA depth in TreeLoRA's performance.

However, I still find some aspects unclear. Specifically, while the explanation about LoRA depth partially clarifies the observed performance drop in LLMs, it remains ambiguous how TreeLoRA itself solves the issue when the model size becomes larger. Furthermore, although the authors state that TreeLoRA's depth is independent of the number of tasks, the rationale for choosing specific depths for different models is still not clearly explained. For instance, in the LLaMA-2-7b-chat experiments, both TreeLoRA and O-LoRA perform poorly at depth 8 but significantly improve at depth 64, raising the question of whether TreeLoRA consistently outperforms O-LoRA or whether its benefits only appear under particular depths. As there is no empirical evidence indicating a linear relationship between performance and LoRA depth, the observed improvements remain difficult to interpret.

The newly added results are appreciated and add value to the submission. Moreover, the idea behind TreeLoRA represents a novel and promising research direction. However, I believe further clarification is needed regarding TreeLoRA’s consistent advantage on LoRA depths. Therefore, I maintain my original score.

**Claims And Evidence:**

The claims made in the submission are supported by clear and convincing evidence.

**Essential References Not Discussed:**

This paper uses the image datasets CIFAR-10 and ViT but it does not compare with the similar work [InfLoRA] published in CVPR 2024, which also uses the same datasets and the same model.

[1] Liang, Yan-Shuo, and Wu-Jun Li. "Inflora: Interference-free low-rank adaptation for continual learning." Proceedings of the IEEE/CVF Conference on Computer Vision and Pattern Recognition. 2024.

**Experimental Designs Or Analyses:**

I checked the experimental designs. Please see the questions.

**Methods And Evaluation Criteria:**

The proposed methods make sense for the problem, but for the benchmark datasets, this paper does not use the same dataset used by O-LoRA, since it is a direct and very related baseline for this paper.

**Other Comments Or Suggestions:**

Include a discussion on the impact of task order on the performance of TreeLoRA.

**Other Strengths And Weaknesses:**

Strengths:
1. TreeLoRA is a novel hierarchical structure that efficiently groups tasks based on gradient similarity, efficient task adaptation, and knowledge sharing.
2. This paper provides a theoretical analysis to support the proposed method’s efficiency, demonstrating regret bounds compared to conventional methods.
3. The proposed method achieves speed improvements while having similar or even better performance compared to current existing methods.

Weaknesses:
1. There is limited analysis of how the performance of TreeLoRA is affected by key hyperparameters such as the tree depth and gradient similarity threshold. The authors mentioned that the tree depth is set to 5 for ViT and 64 for LLMs, but there is insufficient discussion on how the values are chosen and how sensitive the method is to the choices.
2. The authors compare TreeLoRA with O-LoRA and other baselines, but there is insufficient analysis of how different task orderings affect performance. O-LoRA explored the impact of different task orders, it would strengthen the evaluation if using the same task orders to compare.
3. The paper does not use the same datasets as O-LoRA, which makes the current comparison less rigorous and kind of unfair.

**Questions For Authors:**

1. How does TreeLoRA handle situations where task order changes over time? Since the experiments compare with the recent baseline, O-LoRA, which explored the impact of different task orders, how does TreeLoRA perform on the different task orders in O-LoRA.
2. How does TreeLoRA determine the depth of the K-D tree chosen for ViTs (5) and LLMs (64)? Is there any strategy for guidance? Is there any range for the depth? Does it connect with the number of tasks?
3. In the experiment, the TreeLoRA uses image datasets CIFAR-10 as one benchmark, and I found one previous work, InfLoRA (CVPR 2024), which also utilizes this dataset and the same model ViT to conduct the experiments. How does TreeLoRA’s performance compare to this work?

[1] Liang, Yan-Shuo, and Wu-Jun Li. "Inflora: Interference-free low-rank adaptation for continual learning." Proceedings of the IEEE/CVF Conference on Computer Vision and Pattern Recognition. 2024.

**Relation To Broader Scientific Literature:**

As mentioned in the paper, the proposed method may help to decrease energy consumption and carbon emissions associated with training AI models, contributing to environmentally sustainable machine learning.

**Theoretical Claims:**

I checked the correctness of the theorem 1.

---

> ### Author Rebuttal · Authors · 2025-03-31
>
> We sincerely appreciate the reviewer's feedback. In the following, we address each of your technical inquiries.
>
> **Q1.** "impact of task order on the performance ...this paper does not use the same dataset used by O-LoRA."
>
> **A1.** Thank you for your comment. First, we would like to clarify that the TRACE dataset used in our paper consists of 8 tasks, which is ***larger*** than the 4 tasks used in the O-LoRA paper. Moreover, the TRACE dataset includes a ***diverse set*** of tasks, such as text generation and code generation, whereas the datasets used in the O-LoRA paper primarily focus on classification tasks.
>
> To further address your concern, we conduct additional experiments to validate TreeLoRA and other contenders using the _**same datasets**_ (i.e., dbpedia, amazon, yahoo, and agnews) and  _**same task orders**_ as in O-LoRA. We use *Llama-3.2-1B-Instruct* as the foundation model, and convert classification tasks to text generation tasks. Results of overall performance (%)/BWT (%) indicate that TreeLoRA achieves superior performances across different orders, and also improves efficiency (about 1.5x speedup compared to O-LoRA):
>
> Task Order|FIX|OGD|GEM|SeqLoRA|HideLoRA|O-LoRA|TreeLoRA
> :-:|:-:|:-:|:-:|:-:|:-:|:-:|:-:
> Order1|48.75/0.0|54.16/10.82|54.10/10.25|55.71/8.63|56.32/2.75|59.50/2.51|**59.73/2.22**
> Order2|48.75/0.0|46.82/21.50|46.70/20.93|45.26/7.70|53.41/5.58|52.53/5.65|**53.78/5.74**
> Order3|48.75/0.0|43.79/27.31|51.12/16.24|49.03/19.05|61.25/3.12|**63.82/2.03**|62.76/2.23
> Time|-|684|712|43|56|58|**45**|
>
> Further, we include an experiment involving long task sequences, which is similar to the experimental setup used in O-LoRA's paper. For more details, please refer to the **A1** for Reviewer ygph. We will add these results in the revised version. Thanks again for your valuable comments.
>
> ---
>
> **Q2.** There is limited analysis of how the performance of TreeLoRA is affected by key hyperparameters. How does TreeLoRA determine the depth of the K-D tree chosen for ViTs (5) and LLMs (64)? Is there any strategy for guidance? Is there any range for the depth? Does it connect with the number of tasks?
>
> **A2.** Thanks for your comment. We detail the hyperparameter analysis of our method below:
>
> - **Hyperparameter Sensitivity.** In our paper, we provided an analysis of the sensitivity of TreeLoRA's performance to key hyperparameters, such as the regularization coefficient $\lambda$ and the learning rate $\alpha$. These results are detailed in Appendix A.6.
>
> - **Impact of the Tree Depth.** To further explore the impact of tree depth, we conducted additional experiments to validate how varying tree depth affects the model's performance, with overall performance (%) and training time shown in the table below:
>
> Tree Depth|CIFAR-100 (ViT)|Time (s)
> :-:|:-:|:-:
> 1|86.52|171.31
> 2|88.22|182.42
> 5|**88.54**|**212.66**
> 7|88.39|233.17
>
> Tree Depth|TRACE (LLM)|Time (s)
> :-:|:-:|:-:
> 8|21.49|455
> 16|22.62|468
> 32|38.62|476
> 64|**43.52**|**485**
>
> These results show that TreeLoRA is relatively robust to the choice of tree depth. For ViT models, a tree depth of 5 provides a good balance between performance and efficiency, while for LLMs, a depth of 64 is recommended. These settings bring slightly better trade-offs between performance and efficiency.
>
> We clarify that tree depth is not determined by the number of tasks, as a single node in the tree can contain multiple tasks, allowing the structure to scale to a large number of tasks. Additionally, the maximum tree depth should not exceed the number of transformer layers (as illustrated in Figure 1)
>
> - **Impact of the Gradient Similarity Threshold.** Regarding the threshold $\delta$, as mentioned in Section 3.3, we clarify that it is automatically determined and does not need manual adjustment. Specifically, inspired by the K-D tree data structure [Bentley, 1990], at each split, the threshold is computed by taking the median of the similarity (L1-norm) between each task gradient and the mean gradient within the corresponding task group. This approach ensures balanced tree growth and adaptive partitioning of the gradient space, without the need for manual threshold adjustments.
>
> ---
>
> **Q3.** One previous work, InfLoRA (CVPR 2024), also utilizes CIFAR-100 and the same model ViT to conduct the experiments. How does TreeLoRA's performance compare to this work?
>
> **A3.** Thank you for your comment. We add experiments to directly compare the performance with InfLoRA on the CIFAR-100 dataset using ViT models:
>
> ||InfLoRA|TreeLoRA
> :-:|:-:|:-:
> Acc (%)|85.44|**88.54**
> BWT (%)|4.82|**4.37**
> Time (s)|695|**214**
>
> The results demonstrate that TreeLoRA achieves similar accuracy compared to InfLoRA, with lower training times. We will add these results and corresponding discussions in the revised version.
>
> ---
>
> We hope these clarifications address your concerns. We sincerely wish that you can re-evaluate our paper and consider updating the score for our paper. Thank you for your time and feedback!

---

> > ### Comment · Reviewer_bzQv · 2025-04-02
> >
> > Thank the authors for providing additional experiments to clarify in the rebuttal. Based on the authors' responses, I have some further concerns:
> >
> > 1. The answer in Q1 "which is larger than the 4 tasks used in the O-LoRA paper" is a misleading expression since O-LoRA conducted experiments on both 4 tasks in the standard continual learning benchmark and 15 tasks in the large number of tasks.
> >
> > 2. For "Impact of the Tree Depth", it seems like TreeLoRA has better robustness on ViT using CIFAR100 than LLM using TRACE. Since the tree depth has more influence on LLM accuracy and llama-3.2-1B has more parameters than ViT-B/16, it looks like TreeLoRA cannot be simply extended to LLM.

---

> > > ### Author Response · Authors · 2025-04-04
> > >
> > > **[New!] Thanks for your recognition of the novelty of our method and theoretical analysis. We sincerely wish the reviewer could kindly review our newly added experiments, which we believe adequately address all of your concerns.**
> > >
> > > Please do let us know if you have any additional comments (use the "edit" function). With the inclusion of additional experiments and expanded discussions on related work, we'd be deeply grateful if you could consider raising your score to further support our paper.
> > >
> > > ---
> > > We thank the reviewer for the follow-up questions and we address each of your additional concerns in detail.
> > >
> > > **Q1.** "The answer in Q1 'which is larger than the 4 tasks used in the O-LoRA paper' is a misleading expression since O-LoRA conducted experiments on both 4 tasks in the standard continual learning benchmark and 15 tasks in the large number of tasks."
> > >
> > > **A1.** Many thanks for your further comments. We will revise the misleading expression in the next version. We'd like to clarify that our earlier choice of focusing on the standard CL benchmark was based on the following two main considerations:
> > >
> > > -   In the O-LoRA paper, the 15-task benchmark was evaluated using the T5 model only, without including the LLaMA architecture. Since our study focuses on widely-used, decoder-only LLM structures such as LLaMA and Mistral, we prioritized the standard CL benchmark to enable a direct and fair comparison.
> > > -   Moreover, although the 15-task setting includes a greater number of tasks, all of them are classification problems measured by accuracy. As such, the increased quantity may not necessarily suggest greater task diversity or increased difficulty compared to the standard CL benchmark.
> > >
> > > Nonetheless, to more directly address the reviewer's concern, **we have now extended our evaluation to include the full 15-task benchmark, with 3 orders (same as in the O-LoRA paper)**: MNLI, CB, WiC, COPA, QQP, BoolQA, RTE, IMDB, Yelp, Amazon, SST-2, DBpedia, Agnews, MultiRC, and Yahoo, using _meta-llama / Llama-3.2-1B-Instrcut_ as the foundation model. This required additional effort to adapt the new benchmark into our codebase and align it with our pipeline, which has just been completed. The results are as follows:
> > >
> > > |Task Order|FIX|HideLoRA|O-LoRA|TreeLoRA|
> > > |:---:|:-:|:-----:|:---:|:--:|
> > > |Order4|52.13/0.0|59.44/4.33|**59.89/4.67**|58.45/4.98|
> > > |Order5|52.13/0.0|54.49/7.52|57.05/4.42|**58.12/3.31**|
> > > |Order6|52.13/0.0|57.26/6.98|58.02/4.73|**59.00/4.12**|
> > > |Time|-|124|121|83|
> > >
> > > We hope this clarification addresses your concerns. We will continue expanding experiments on these benchmarks using additional foundation models and will report comprehensive results in the revised version of the paper. Thanks!
> > >
> > > ---
> > >
> > > **Q2.** "For 'Impact of the Tree Depth', it seems like TreeLoRA has better robustness on ViT using CIFAR100 than LLM using TRACE..."
> > >
> > > **A2.** We appreciate your insightful observation. We would like to take this opportunity to clarify this phenomenon and provide additional empirical evidence to support the scalability of TreeLoRA to LLMs.
> > >
> > > -   As illustrated in Figure 1 of the main paper, the tree depth in our method design is directly constrained by the **LoRA depth**, i.e., the number of layers where LoRA adapters are applied. LLMs such as LLaMA-3.2-1B or LLaMA-2-7B have significantly more layers and parameters than ViT-B/16, which naturally calls for more LoRA adapters. A shallow LoRA depth (aka tree depth) in such architectures can lead to a performance drop. Therefore, we clarify that **the performance drop should not be attributed to the TreeLoRA architecture itself, but rather to the insufficient LoRA depth**.
> > >
> > > -   To support this interpretation, we conducted an additional experiment comparing with O-LoRA, a widely-acknowledged method in the field, to directly show the influence of constraining LoRA depth, both using _meta-llama / LLaMA-2-7B-Chat_ as the foundation model:
> > >
> > > |LoRA Depth|O-LoRA|TreeLoRA|
> > > |:---:|:----:|:----:|
> > > |8|21.43|21.49|
> > > |64|42.78|43.52|
> > >
> > > As the table shows, both O-LoRA and TreeLoRA exhibit substantial performance drops when the LoRA depth is limited (e.g., depth = 8). When the LoRA depth increases (e.g., depth = 64), TreeLoRA performs comparably—or even slightly better—than O-LoRA. This suggests that TreeLoRA is indeed extensible to LLMs. In practice, the default LoRA depth used in other methods often suffices — often set as the number of layers in the LLM.
> > >
> > > We will incorporate this discussion along with the experimental results into the revised version of the paper to provide a clearer picture of TreeLoRA's scalability.
> > >
> > > ---
> > >
> > > **In summary, we have**
> > >
> > > -   Extended our evaluation to include the full 15-task benchmark, with 3 orders (same as in the O-LoRA paper).
> > > -   Provided a detailed analysis of the relationship between LoRA depth and performance, demonstrating that TreeLoRA can easily scale to large models such as LLMs.

---

### Official Review · Reviewer_GZLm · 2025-03-15

**Overall Recommendation:** 3

**Summary:**

This paper proposes TreeLoRA, a continuous learning method that builds hierarchical adapters based on gradient similarity, which aims to solve the computational efficiency problem in continuous learning of large pre-trained models (LPMs). By organizing tasks into a K-D tree structure and introducing sparse gradient updates, this method achieves better accuracy than baselines (such as HiDeLoRA) on ViT and LLM, while reducing training time by about 2.4 times. However, the core dynamic update mechanism (such as node addition and reduction rules) is not fully explained, and the complexity comparison with mainstream parameter efficient fine-tuning methods (such as LoRA/O-LoRA) is insufficient, which may affect the credibility of the method.

**Claims And Evidence:**

The tree structure can dynamically capture task similarity and reduce computational complexity.
It does not explain how nodes are dynamically increased or decreased (such as the conditions that trigger splitting). The rationality of the structure is only indirectly proved through visualization, and there is a lack of mathematical description of the dynamic process.

**Essential References Not Discussed:**

Rusu et al. (2016) Progressive Neural Networks
Wang et al. (2023) DyLoRA: Dynamic Low-Rank Adaptation

**Experimental Designs Or Analyses:**

The evolution of tree structure under dynamic task flow (such as the tree changes when 10 new categories are added in Split CIFAR-100)
Average adapter parameters per task vs. standard LoRA.
The impact of different tree depth/width on performance.

**Methods And Evaluation Criteria:**

- The construction of tree structure depends on gradient similarity, but it is not clear:
How to update the tree level when new tasks are inserted (incremental or global reconstruction)
Threshold design for node splitting/merging
Adaptive relationship between tree depth and number/similarity of tasks

- Only CL methods such as HiDeLoRA are compared, and no theoretical/experimental comparison is performed with standard LoRA (independent adapter for each task) or O-LoRA (orthogonal constraint adapter), which cannot prove its advantages over basic methods.

**Other Comments Or Suggestions:**

Add parameter/FLOPs comparison experiment with standard LoRA/O-LoRA
Discuss the additional overhead of tree structure maintenance (such as gradient similarity calculation complexity)

**Other Strengths And Weaknesses:**

\

**Questions For Authors:**

Is node splitting based on a fixed threshold? How to avoid over-complication of the tree structure?
When the new task has low similarity with the existing node gradient, how to expand the tree structure? (Add new branches or increase the layer depth?)
Is the theoretical relationship between the number of tree levels L and the number of tasks T O(log T)? How to set L in actual experiments?
Compared with the O(d) parameter increment of O-LoRA (d is the adapter dimension), is the parameter growth rate of TreeLoRA strictly lower?
Is the sparsity rate of sparse gradient updates related to the tree structure? How to balance sparsity and knowledge retention?

**Relation To Broader Scientific Literature:**

\

**Theoretical Claims:**

The regret bound analysis does not consider the adjustment cost of the dynamic tree structure. The theoretical model assumes a static task relationship, which is inconsistent with the actual dynamic scenario.

---

> ### Author Rebuttal · Authors · 2025-03-31
>
> Thanks for your helpful comments! Below, we address your major technical questions and will revise the paper to improve clarity and resolve any potential misunderstandings.
>
> ---
>
> **Q1.** Elaborate more on the construction of the tree structure, including update and expansion, threshold design, relationship between tree depth and number/similarity of tasks.
>
> **A1.** Thanks for the question. The construction of the tree structure is explained in detail below:
>
> - **Update of Tree Structure.** After each task, we store the task-specific LoRA adapter (as in Section 3.3) and update the tree by inserting adapter into the leaf node of nearest branch by a depth-first search (DFS), thus adding new nodes and expanding the tree. If nodes exceed the storage budget, we choose the closest adapters and reduce them to a single one (as in Appendix A.3).
>
> - **Threshold Design.** Inspired by the K-D tree structure [Bentley, 1990], the threshold $\delta$ does not require manual tuning. Specifically, at each split, the threshold is computed by taking the median of the similarity (L1-norm) between each task gradient and the mean gradient within the corresponding task group, ensuring balanced tree growth and adaptive partitioning.
>
> - **Relationship Between Tree Depth and Number/Similarity of Tasks.** We clarify that tree depth is not directly determined by the number of tasks, as a single node can contain multiple tasks, allowing it to scale to a large number of tasks. However, the depth should not exceed the number of transformer layers (as illustrated in Fig. 1), and it is treated as a tunable hyperparameter. Further empirical analysis of tree depth is provided in **A2** in response to Reviewer bzQv.
>
> We will add these details in the revision for more clarity.
>
> ---
>
> **Q2.** No comparison with standard LoRA or O-LoRA.
>
> **A2.** We believe this is a misunderstanding here — we have compared our approach with the stand LoRA (aka SeqLoRA) and O-LoRA in our experiments, as presented in Table 3 of the submission PDF. The results demonstrate our improvements over basic methods in both performance and efficiency. We also add FLOPs comparison with LoRA/O-LoRA, please refer to **A5**.
>
> ---
>
> **Q3.** On Theoretical Claims "does not consider the adjustment cost of the dynamic tree structure... assumes a static task relationship"
>
> **A3.** Thank you for the insightful comments. Our regret analysis focuses on a simplified scenario to provide foundational justifications for the proposed algorithm. The elements you mentioned can certainly be incorporated into future work by some modern online learning techniques. For instance, incorporating the *switching cost* would allow us to account for the cost of adjusting the tree structure, and adopting *dynamic regret* could help capture time-varying task relationships. Nonetheless, we believe these extensions are non-trivial to achieve. For instance, the minimax rate for MAB is $\Theta(\sqrt{T})$, whereas introducing a switching cost increases it to $\Theta(T^{2/3})$ [Dekel et al., STOC'14]. Our theoretical results serve as a first step toward tackling more complex scenarios, and we will include these points in the discussion of future work.
>
> ---
>
> **Q4.** Theoretical relationship.
>
> **A4.** In Theorem 1, the regret bound is dependent on both the number of tasks $N$ and the task complexity $J_n$ (which is controlled by the similarity between tasks). Consequently, as the number of tasks increases or as tasks become less similar, the performance of our method is expected to deteriorate, requiring more rounds to maintain and search the tree structure.
>
> ---
>
> **Q5.** Other concerns about experiments, including:
>
> - [5-1] parameter/FLOPs comparison, additional overhead of tree structure, "is the parameter growth rate of TreeLoRA strictly lower (than O-LoRA)"
> - [5-2] evolution of tree structure under dynamic task flow
> - [5-3] impact of different tree depths on performance.
>
> **A5.** Thanks for your suggestions.
>
> - [A5-1] We include a comparison using *LLaMA-2-7B-Chat* as an example for training a single token on the 10-th task, as in table below:
>
> ||FLOPs|Parameter Complexity
> -|-|:-:
> OGD|28×10⁹|$\mathcal{O}(mn)$
> LoRA|4.2×10⁶|$\mathcal{O}((m+n)r)$
> O-LoRA|4.2×10⁷|$\mathcal{O}((m+n)rN)$
> TreeLoRA|4.2×10⁶|$\mathcal{O}((m+n)r+Nr)$
>
> Here, $m$ and $n$ denote the dimensions of the transformer's parameter matrix, $r$ is LoRA rank, and $N$ is the number of tasks.
>
> - [A5-2] We also add an analysis of the evolution of the tree structure under dynamic task flow, which provides a clearer illustration of how TreeLoRA captures task structures over time: https://anonymous.4open.science/r/TreeLoRA/scripts/rebuttal.jpg
>
> - [A5-3] Additionally, we conduct further experiments to evaluate the impact of tree depth on performance, please refer to **A2** for Reviewer bzQv.
>
> ---
>
> Thank you again for the helpful review. We will revise the paper accordingly and include the related references, such as Rusu et al. (2016) and Wang et al. (2023).

---

### Decision · Program_Chairs · 2025-05-01

**Decision:**

Accept (poster)

**Comment:**

This paper introduces a hierarchical, gradient-similarity-based adapter structure (TreeLoRA) for continual learning with large pre-trained models. The main contributions include:
- A novel hierarchical adapter structure that organizes task-specific LoRAs via a K-D tree guided by layer-wise gradient similarity, enabling scalable and efficient parameter reuse.
- A bandit-based task similarity estimation mechanism and sparse gradient updates that improve learning efficiency in large vision and language models.
- Extensive empirical validation across ViT and LLM benchmarks, showing improved performance and computational efficiency over strong baselines (e.g., O-LoRA, InfLoRA), alongside theoretical regret bounds supporting the algorithm’s design.

During the response period, the authors thoroughly addressed reviewer concerns, including comparisons with InfLoRA, SAPT, and TASL, ablations on tree depth, task order sensitivity, and LoRA depth. They also added experiments on long task sequences and 13B-scale LLMs. The authors also provided experiments in response to Reviewer bzQv's only remaining concern.

Given the paper’s clear motivation, and strong empirical results, I recommend acceptance. For the final version, the authors are encouraged to (1) further clarify how LoRA depth interacts with model architecture in scaling to LLMs, and (2) more explicitly highlight comparisons with non-LoRA-based baselines and plug-in strategies (e.g., replay, generative modules).